# Father's adolescent body silhouette is associated with offspring asthma, lung function and BMI through DNA methylation

Negusse Tadesse Kitaba [1], Toril Mørkve Østergaard[2], Marianne Lønnebotn[3], Simone Accordini[4], Francisco Gómez Real[5], Andrei Malinovschi[6], Anna Oudin[7], Bryndis Benediktsdottir[8], Francisco Javier Callejas González[9], Leopoldo Palacios Gómez[10], Mathias Holm[11], Nils Oskar Jõgi[6], Shyamali C. Dharmage[12], Svein Magne Skulstad[2], Vivi Schlünssen[13], Cecilie Svanes [2,14,16] ✉ & John W. Holloway [1,15,16]

Boys' pubertal overweight associates with future offspring's asthma and low lung function. To identify how paternal overweight is associated with offspring's DNA methylation (DNAm), we conducted an epigenome-wide association study of father's body silhouette (FBS) at three timepoints (age 8, voice break and 30) and change in FBS between these times, with offspring DNAm, in the RHINESSA cohort (N = 339). We identified 2005 differentially methylated cytosine-phosphate-guanine (dmCpG) sites (FDR < 0.05), including dmCpGs associated with offspring asthma (119), lung function (178) and BMI (291). Voice break FBS associated with dmCpGs in loci including *KCNJ10, FERMT1, NCK2* and *WWP1*. Change in FBS across sexual maturation associated with DNAm at loci including *NOP10, TRRAP, EFHD1, MRPL17* and *NORD59A;ATP5B* and showed strong correlation in reduced gene expression in loci *NAP1L5, ATP5B, ZNF695, ZNF600, VTRNA2-1, SOAT2* and *AGPAT2*. We identified 24 imprinted genes including: *VTRNA2-1, BLCAP, WT1, NAP1L5* and *PTPRN2*. Identified pathways relate to lipid and glucose metabolism and adipogenesis. Father's overweight at puberty and during reproductive maturation was strongly associated with offspring DNA, suggesting a key role for epigenetic mechanisms in intergenerational transfer from father to offspring in humans. The results support an important vulnerability window in male puberty for future offspring health.

Even though the impact of obesity on the individual is clearly understood, the prevalence of childhood obesity is increasing across the globe[1]. Recent epidemiological studies from the RHINESSA/RHINE/ECRHS cohorts and the Tasmanian Longitudinal Health Study (TAHS) have shown that overweight in boys at the age of voice break (puberty) may impair not only their own health but also the health of their future offspring, in terms of higher asthma risk, lower attained stature and lower lung function[2–4].

An exposure-sensitive period during prepubertal years has also been highlighted by observations from the Överkalix and ALSPAC cohorts, where excess food supply and smoking in mid-childhood have been linked to metabolic and cardiovascular health, and risk of obesity in subsequent generation(s)[5–7]. Father's preconception of overweight/obesity has further been associated with offspring's obesity and altered metabolism[8]. Supporting a role for intergenerational epigenetic mechanisms in humans, Kitaba et al.[9] found that fathers' exposure to tobacco smoke, particularly during adolescent years, was associated with altered epigenetic patterns in their future offspring.

Efforts in identifying molecular mechanisms underlying these findings have suggested a pre-conceptional influence on epigenetic (re)programming during germ cell development, where the sperm epigenome is increasingly plastic and responsive to environmental exposures which can possibly affect epigenetic states and give rise to pleiotropic effects in future offspring if transmitted to the next generation at fertilisation[10–14].

Male obesity has been found to alter spermatocyte DNA methylation patterns[15–17] and non-coding tRNA content[18], as well as seminal plasma composition[19,20]. Offspring of obese fathers have also been shown to display altered DNA methylation levels at several regulatory regions of imprinted genes[21,22]. Given that fathers' BMI around conception has also been demonstrated to be an independent determinant of offspring metabolic health[18] as well as to be associated with offspring birthweight and epigenome-wide methylation patterns up to the age of 7[23], this clearly supports the hypothesis that BMI-related differences in sperm content can be transmitted to subsequent offspring and impact on their health and development.

The present study aimed to investigate whether fathers' body silhouettes (FBS) (as a surrogate measure of body composition) and their trajectory across adolescence and young adulthood were associated with offspring DNA methylation patterns and whether identified signals were associated with offspring phenotypic outcomes in terms of asthma, lung function and BMI. We hypothesised that differential DNA methylation patterns in offspring might reflect the molecular mechanisms underlying the effects of fathers' obesity in adolescence on offspring health observed in epidemiological studies[2–4]. We have previously shown that the use of self-reported figural body silhouettes provides a valid tool for assessing overweight and obesity retrospectively[24]. In a two-generation cohort, we sought to identify the DNA methylation pattern in whole blood of offspring (aged 7–51 years) associated with (1) the father's preconception body silhouette at ages 8, voice break and 30 years and (2) trajectory of father's body silhouette across adolescence and young adulthood measured as change from age 8 years to voice break and from voice break to age 30. As previous epidemiological studies have reported sex-specific health outcomes of paternal obesity on offspring[2], we also wanted to explore whether patterns of association between fathers' preconception body silhouette and offspring DNA methylation were different between sons and daughters.

## Methods

### Study design and data
Data and samples from offspring were available from the RHINESSA study (www.rhinessa.net)[25]. Parental data was retrieved from the population-based European Community Respiratory Health Survey (ECRHS, www.ecrhs.org) and/or the Respiratory Health in Northern Europe (RHINE, www.rhine.nu) studies. In this study, 339 offspring-father pairs with complete data on fathers' body silhouettes and offspring DNA methylation were included (Table 1). These participants, all of the white European ethnicity, were from six study centres (Aarhus, Denmark; Albacete/Huelva, Spain; Bergen, Norway; Melbourne, Australia; Tartu, Estonia). Medical research committees in each study centre approved the study and all participants gave written consent Ethical permissions were obtained for each study wave from the local ethics committee in each of the participating centres. Details of the ethics committees and approval reference numbers for each study centre are listed on www.rhinessa.net. All ethical regulations relevant to human research participants were followed.

### Definitions of father's body silhouettes (FBS) before conception
Father's body silhouette (FBS) was measured using a validated figural drawing scale of 9 sex-specific body silhouettes at each time point (age 8, voice break, 30 years)[24]. The figural drawing scale has been validated against self-reported height and weight both for current and past body silhouettes[24]. We applied a cut-off of Fig. 5 or greater to classify fathers as being overweight. This is the same cut-off that a previous validation study identified as optimal for identifying overweight people (BMI, 25–30 kg/m$^2$)[26]. Change in the father's body silhouette between age 8 and voice break, and voice break and age 30 were defined according to changes in body silhouette figure between the two time points. Aspects of change were also investigated according to whether they switched from being normal weight to overweight or vice versa across adolescence and young adulthood, and according to how many body figures gain or loss in body silhouettes spanned (see Supplementary Fig. 1).

### Offspring outcomes
Adult offspring ever having asthma was based on answers to the question: "Have you ever had asthma diagnosed by a doctor?". Lung function (forced expiratory volume in 1 s (FEV1) and forced vital capacity (FVC)) was measured at RHINESSA clinical examinations as previously described[27].

### Methylation profiling and processing
DNA methylation in offspring was measured in DNA extracted from peripheral blood, using a simple salting out procedure[28]. Bisulfite-conversion was undertaken using EZ 96-DNA methylation kits (Zymo Research, Irvine, CA, USA) at the Oxford Genomics Centre (Oxford, UK), and methylation was assessed using Illumina Infinium MethylationEPIC Beadchip arrays (Illumina, Inc., CA, USA) with samples randomly distributed on microarrays to control against batch effects.

### DNA methylation quality control and normalisation
Data analysis was undertaken using R v 4.1[29] and methylation quality was assessed using minfi[30] and Mefil[31]. To remove technical variation detected by SVD plot[32] using champ[33], combat from SVA was applied for both batch and slide variables[34]. Cell-type proportions were estimated using Epigenetics Dissection of Intra-Sample Heterogeneity (EpiDISH)[35]. Normalisation was carried out using BMIQ[36]. Probes were excluded from analysis using the following criteria: Detection $p$ above 0.01, probes with a beadcount <3 in at least 5% of samples ($N = 1357$), non-cg probes ($N = 2762$), SNPs as identified by Zhou[37] ($N = 93,900$), aligned to multiple locations as identified in Nordlund[38] ($N = 15$), probes on the X or Y chromosomes (16,089), and cross-reactive probes on the EPIC array (2382)[39]. A total of 730,820 probes were used for downstream analysis.

### Statistical analysis
To identify differentially methylated Cytosine-phosphate-Guanine sites (dmCpGs), the associations of offspring DNA methylation beta-value as outcome (continuous outcomes) with paternal body silhouette phenotypes (normal or overweight/obese, independent variables) were assessed using robust linear regression models using limma[40]. Covariates included offspring age, sex, estimated blood cell proportions (B-cells, Natural Killer cells, CD4 T-cells, CD8 T-cells, Monocyte, Neutrophils) and socioeconomic class. Eosinophils were not included due to a very low estimate and to avoid potential multicollinearity[41]. Grandparents' education was used as a proxy for paternal social class in childhood, either as low or high. Social class was assigned as high for university-level education or if both grandparents attended high school, otherwise it was assigned as low. Multiple test correction was applied using the Benjamini and Hochberg method[42] where a false discovery rate (FDR) corrected $p \leq 0.05$ was considered as statistically significant in the covariates corrected regression model.

### The FBS-CpG associations were assessed by running

- EWAS analyses at each of the paternal age time points: Father body silhouette at age 8 (FBS-8), Father body silhouette at voice break (FBS-V), and at Father body silhouette at age 30 (FBS-30).
- A sex-stratified EWAS analysis of Father body silhouette at voice break in females (FBS-Vf) and Father body silhouette at voice break in males (FBS-Vm) to investigate whether epigenetic signals differ between male and female offspring.
- EWAS analyses of change in paternal body silhouettes based on both ordinal (Father body silhouette change between age 8 and voice break (FBS-V8c), Father body silhouette change between voice break and age 30 (FBS-V30c) and categorical scale measures of gradations in body size (Father body silhouette gain or reduced between age 8 and voice break (FBS-V8gr)/Father body silhouette gain or reduced between voice break and age 30 (FBS-V30gr) and Father body silhouette retain or swap between age 8 and voice break (FBS-V8rs)/ Father body silhouette retain or swap between voice break and 30 (FBS-V30rs) to

**Table 1 | Characteristics of study participants according to father's body silhouette (FBS) at age 8 years, at voice break and at age 30 years**

| Variable | A: FBS 8[b] Normal N=304[a] | Overweight N=34[a] | p[b] | B: FBS Voice break[b] Normal N=307[a] | Overweight N=32[a] | p[b] | C: FBS 30[c] Normal N=288[a] | Overweight N=57[a] | p[b] | D: FBS Voice break by sex[d] Normal | Overweight | p[c] | E: Offspring age[d] Normal | Overweight | p[d] |
|---|---|---|---|---|---|---|---|---|---|---|---|---|---|---|---|
| **Offspring sex** | | | 0.4 | | | 0.4 | | | 0.4 | | | 0.4 | | | 0.12 |
| F | 147 (48%) | 19 (56%) | | 147 (48%) | 18 (56%) | | 138 (48%) | 31 (54%) | | 147 (89%) | 18 (11%) | | 25 (21, 32) | 27 (21, 35) | |
| M | 157 (52%) | 15 (44%) | | 160 (52%) | 14 (44%) | | 150 (52%) | 26 (46%) | | 160 (92%) | 14 (8.0%) | | | | |
| **F: Social class[e]** | | | 0.2 | | | 0.9 | | | 0.039 | | | | | | |
| High | 78 (26%) | 5 (15%) | | 78 (25%) | 7 (22%) | | 65 (23%) | 22 (39%) | | | | | | | |
| Low | 122 (40%) | 13 (38%) | | 122 (40%) | 14 (44%) | | 119 (41%) | 19 (33%) | | | | | | | |
| Unknown | 104 (34%) | 16 (47%) | | 107 (35%) | 11 (34%) | | 104 (36%) | 16 (28%) | | | | | | | |

| Variable | F | M | p |
|---|---|---|---|
| **G: Offspring ever had asthma[e]** | 33 (20%) | 34 (20%) | >0.9 |
| **H: Offspring BMI[f]** | 22.4 (20.6, 24.6) | 25.0 (23.1, 27.9) | <0.001 |
| **I: Offspring current body weight[g]** | | | <0.001 |
| Normal | 317 (92%) | 234 (62%) | |
| Overweight | 28 (8.1%) | 145 (38%) | |

| J: Offspring lung function[h] | FVC_pre* | FEV1_pre* | FEV1_FVC_pre* | FVC_post* | FEV1_post* | FEV1_FVC_post* |
|---|---|---|---|---|---|---|
| F | 3.87 (3.50, 4.24) | 3.24 (2.97, 3.57) | 0.85 (0.80, 0.88) | 3.85 (3.54, 4.24) | 3.34 (3.04, 3.68) | 0.87 (0.84, 0.90) |
| M | 5.43 (4.83, 5.97) | 4.43 (4.01, 4.82) | 0.82 (0.78, 0.85) | 5.30 (4.88, 5.90) | 4.59 (4.12, 4.93) | 0.84 (0.82, 0.88) |

*FEV1_pre* pre-bronchodilator forced expiratory volume in 1 s, *FVC_post* forced vital capacity post-bronchodilator, *FEV1_post* FEV1 post-bronchodilator, *FEV1_FVC_post* FEV1/FVC post-bronchodilator.

*p < 0.001.

[a] n (%).

[b] Pearson's Chi-squared test.

[c] n (%), Median (Q1, Q3).

[d] Pearson's Chi-squared test; Wilcoxon rank sum test.

[e] Median (Q1, Q3), n (%).

[f] Wilcoxon rank sum test, Pearson's Chi-squared test.

[g] Median (Q1, Q3).

[h] Wilcoxon rank sum test.

**Table 2 | Characteristics of offspring for Current BMI, age and sex by study centre (N = 724)**

| Offspring | Albacete N = 48[a] | Arhus N = 48[a] | Bergen N = 395[a] | Huelva N = 29[a] | Melbourne N = 75[a] | Tartu N = 129[a] | p-value[b] |
|---|---|---|---|---|---|---|---|
| Sex |  |  |  |  |  |  | 0.7 |
| Female | 21 (44%) | 26 (54%) | 180 (46%) | 15 (52%) | 40 (53%) | 63 (49%) |  |
| Male | 27 (56%) | 22 (46%) | 215 (54%) | 14 (48%) | 35 (47%) | 66 (51%) |  |
| Age | 30 (25, 38) | 29 (23, 37) | 27 (23, 33) | 36 (25, 39) | 31 (25, 37) | 30 (25, 34) | <0.001 |
| BMI | 23.1 (21.0, 26.0) | 22.9 (21.3, 25.8) | 23.9 (21.9, 26.6) | 23.1 (20.4, 27.5) | 23.4 (21.1, 25.8) | 23.8 (21.2, 27.0) | 0.3 |

[a]n (%); Median (Q1, Q3).
[b]Pearson's Chi-squared test; Kruskal–Wallis rank sum test.

explore effects of change in FBS across adolescence/young adulthood on the offspring epigenome.

- An EWAS analysis to assess the comparability of offspring's current BMI and body silhouette.

## Downstream enrichment analysis and biological interpretation of dmCpGs

Manhattan plots were generated using qqman[43]. Inflation from systematic biases (measured by genomic factor lambda $\lambda$) was adjusted using BACON[44]. Differentially methylated regions were detected using DMRcate[45] and dmrff[46]. Transcription factor binding site prediction was performed using eFORGE TF[47]. Gene–disease phenotype associations were identified using Open Targets[48]. The EWAS atlas[49] was used to assess dmCpGs for association with known biological traits and the effect of methylation on gene expression. For dmCpGs mapped to genes, gene function ontology (GO) terms were identified using String[50] and enrichr[51]. KEGG pathways were generated and visualised using Cytoscape[52,53]. The methylGSA R-package which accounts for the representation of probes per gene on the EPIC array was used to test for GO enrichment[54]. DMR regional enrichment was carried out using the goregion function from missMethyl R package[55]. Genes overlapping with the GWAS catalogue gene set for obesity traits were identified using FUMAGWAS GENEN2FUNC[56] and dbGAP[51]. Lookup for the association of SNPs with methylation to identify methylation quantitative trait loci (mQTL) was carried out using the goDMC[57] and the MeQTL EPIC[58] databases. The look-up for overlap of known human imprinting genes used reference imprinting genes from https://geneimprint.com/site/genes-by-species.Homo+sapiens. Over-representation of metastable epialleles was compared with CpGs identified by Silver et al.[59].

## Association dmCpGs and offspring outcomes

The identified dmCpG sites across all EWAS were analysed for association with offspring clinical phenotypic traits, specifically BMI, asthma and lung function (FEV1 and FVC) using linear and logistic regression.

## Reporting summary

Further information on research design is available in the Nature Portfolio Reporting Summary linked to this article.

## Results

The present analyses included 339 RHINESSA offspring (Table 1), 174 males and 165 females, aged 7–50 years. Of these, 307 had a father who reported a normal body silhouette at voice break, and 32 of the offspring had a father who reported an overweight/obese body silhouette at voice break. There was a significant sex difference between males and females for BMI and lung function. Offspring current BMI, sex and age distribution by study centre are shown in Table 2.

## Associations of father body silhouette (FBS) with offspring DNA methylation

The summary statistics and sample size for each EWAS are shown in Table 3. Of the three specific preconception age windows, father's body silhouette at voice break was associated with a larger number of differentially

methylated CpGs in offspring compared to father's body silhouette in childhood or young adulthood. Sex-specific EWAS analyses identified more dmCpGs related to FBS at voice break in female than in male offspring. In particular, EWAS exploring changes in father's body silhouette figures across adolescence, i.e. from a normal weight to an overweight body silhouette status or vice versa, were associated with altered DNA methylation patterns in the offspring. In total, we identified 2005 (1962 unique) dmCpGs that showed association with father's normal versus overweight/obese body silhouette before conception (FDR corrected $p \leq 0.05$). Key results for each EWAS are described below; the full list of dmCpGs and adjusted $p$ for each EWAS are provided in Supplementary Data 1A.

Father's body silhouette at age 8 (FBS-8): We identified 12 dmCpG sites associated with father's normal versus overweight/obese body silhouette at age 8 (FDR < 0.05). These sites were mapped to 7 protein-coding genes and 6 intergenic regions. Eight were hypermethylated and 4 hypomethylated with association coefficients ranging from −0.04 to 0.03. Eight of the sites were in the open sea (Supplementary Data 1A: FBS-8).

Father's body silhouette at voice break (FBS-V, FBS-Vm, FBS-Vf): We identified 41 dmCpG sites associated with the father's body silhouette at voice break; these mapped to 32 genes (29 protein-coding and 3 noncoding RNA) and 11 intergenic regions as shown in Table 4. The majority of dmCpGs were hypermethylated in the overweight group (71%) and, relative to CpG Islands, 66% were located in the open sea. For several of the top 10 dmCpGs (FDR corrected $p \leq 0.05$), the methylation distribution showed a pattern of increasing or decreasing methylation according to increasing overweight in the father (increasing levels of FBS) (Fig. 1).

In male offspring, we identified 32 dmCpGs mapped to 26 genes and 6 intergenic regions. 16 were located in open sea (see Supplementary Data 1A: FBS-Vm). In female offspring, we identified 370 dmCpG sites. Of these, 248 dmCpGs were mapped to coding genes and 122 to intergenic regions (Supplementary Data 1A: FBS-Vf).

We identified 6 genomic regions that were differentially methylated in offspring whose fathers had a normal versus overweight/obese FBS status at voice break. Using DMRcate differentially methylated regional enrichment analysis at voice break identified a DMR in SLC44A4 (8 sites) which was enriched for choline metabolism in cancer, glycolysis/gluconeogenesis, citrate cycle (TCA cycle) and fatty acid metabolism (see Supplementary Data 2: DMR:FBS-V DMRcate). In the analysis stratified by offspring sex, we detected 1 DMR in males and 15 DMRs in females. The female DMR sites included ADAMTS16, RNASE1 and AGAP2 at dmr.p.adjust < 0.05 (Supplementary Data 2).

To identify any overlap between the methylation of dmCpGs and known transcription factor (TF) binding sites, we interrogated eForgeTF by selecting CD34_T0 cell line, at https://eforge-tf.altiusinstitute.org (accessed on 23 August 2023). In the FBS analyses at voice break, we found 17 dmCpGs overlapping with 28 TF binding sites (q-value < 0.05). The methylation of cg25020933 (B4GALNT4) overlapped with 6 TFs, cg24420089 (PTDSS2) overlapped with 4 TFs and both cg05509659 (ROBO3) and cg09655253 (MOCS1) overlapped with 2 TFs (Supplementary Data 3). In sex-stratified analyses of FBS at voice break, we found overlapping with 30 TFs (q-value < 0.05) in male offspring and with 6 TFs (q-value < 0.05) in females (Supplementary Data 3).

**Table 3 | Summary statistics of dmCpGs for all EWAS at FDR ≤ 0.05**

| EWAS on father body silhouette | Label | N dmCpGs | Inflation | N persons |
|---|---|---|---|---|
| Father body silhouette at age 8 years | FBS-8 | 12 | 1.2 | 338 |
| Father body silhouette at voice break | FBS-V | 41 | 1.3 | 339 |
| Father body silhouette at voice break in male | FBS-Vm | 32 | 1.291 | 174 |
| Father body silhouette at voice break in female | FBS-Vf | 370 | 1.4 | 165 |
| Father body silhouette at age 30 years | FBS-30 | 3 | 1.12 | 345 |
| Father body silhouette change between voice break and age 8 | FBS-V8c | 277 | 1.17 | 334 |
| Father body silhouette gain or reduced between voice break and age 30 | FBS-V8gr | 222 | 1.2 | 334 |
| Father body silhouette retain or swap between age 8 and voice break | FBS-V8rs | 176 | 1.05 | 334 |
| Father body silhouette change between voice break and age 30 | FBS-V30c | 791 | 1.176 | 334 |
| Father body silhouette gain or reduced between age 8 and voice break | FBS-V30gr | 69 | 1.4 | 334 |
| Father body silhouette retain or swap between voice break and age 30 | FBS-V30rs | 12 | 1.003 | 334 |
| | Total | 2005 | | |

*dmCpG* differentially methylated cytosine-phosphate-guanine site, FDR < 0.05.

In dmCpGs associated with FBS at voice break, sites located to *ZNF570, ZNF569, B4GALNT4, LIPG, PTDSS2* and *GATA5* showed correlation with gene expression (Supplementary Data 4: FBS-V). In the female-stratified analyses of FBS at voice break (*n* = 123), several of the identified dmCpGs showed correlation with gene expression and mapped to genes including *MAGI2, GALNT9, SLCO3A1, OR2I1P, GPR153, TBC1D2B, TNFRSF1, NCAM1* and *JAG1* (Supplementary Data 4: FBS-Vf).

The summary of the number of dmCpGs that showed correlation with gene expression and full details for both promoter and gene-body associated dmCpGs across the six tissues are provided in Supplementary Data 4.

Pathway analysis: Enrichr was used to assess the enrichment of curated signalling pathways among identified dmCpGs. For FBS at voice break-related genes, 31 GO terms (*p* < 0.05) were identified including gastrin signalling pathway (*FOXO3; EGFR; CD44*), molybdenum cofactor (Moco) biosynthesis (*MOCS1*), insulin-signalling in adipocytes (*TBC1D4*), NAD metabolism, sirtuns and aging (*FOXO3*), metabolic pathway of LDL, HDL and TG (*LIPG*), glycerolipids and glycerophospholipids (*PTDSS2*) and ferroptosis (*LPCAT3*). Using methylRR package GSEA we identified GO terms including cellular response to lipopolysaccharide (GO:0071222) and with AGSEA-Promotor1, cellular lipid catabolic process (GO:0044242) (Supplementary Data 5).

Father's body silhouette at age 30 years (FBS-30): We identified three dmCpGs mapping to three genes, *QSOX2, DLGAP2* and *PCDHG* (Supplementary Data 1A: FBS-30).

Change in father's body silhouette between age 8 and voice break: A total of 277 dmCpGs showed association with change in FBS between age 8 and voice break (*λ* = 1.17) (Supplementary Data 1: FBS-V8c). The top hit, cg20668887, mapped to *NOP10* (FDR = 1.68e−91); the association coefficients ranged from −0.34 to 0.42 (Figs. 2, 3). Some genomic loci had multiple dmCpGs such as *NFYA;LOC22144, PTPRN2* and *NAP1L5;HERC3* with the majority (81%) of dmCpGs showing hypermethylation. In the EWAS model using categorical cut-offs of FBS change, we identified 222 dmCpGs at FDR ≤ 0.05 associated with father´s gain or loss in FBS (Supplementary Data 1: FBS-V8gr). We detected 176 dmCpGs (*λ* = 1.05) associated with father´s switch in normal or overweight/obese FBS status between age 8 and voice break. The top hit, cg10157663, mapped to *CCDC178* (FDR = 3.91e−21) with 78% of dmCpGs showing hypermethylation. Many loci were represented by more than 1 dmCpG including *PTCH1* (10 sites), *GABRG1* (7 sites), *BLCAP* (6 sites), *SORCS1* and *HIST1H2BE* (3 sites) (Supplementary Data 1A: FBS-V8rs).

Change in father's body silhouette between voice break and age 30: We identified 791 dmCpGs associated with a change in FBS between voice break and age 30 years; 559 were mapped to coding genes (Supplementary Data 1: FBS-V30c). The coefficient of association ranged from −0.53 to 0.82, as

shown by the volcano plot, where 68% showed hypermethylation (Fig. 2). The genome-wide distribution is shown in Fig. 3. The top hit, cg18950772, mapped to an intergenic region (FDR = 6.23e−261) followed by *SLC25A10* (FDR = 3.21e−205). Some genes were represented by many dmCpGs including *NUP210L* and *TRPM4, WT1, MIR886, NFYA;LOC221442, DIP2C* and *AGPAT2*. In the EWAS models investigating categorial classifications of change in FBS from voice break to age 30, we identified 69 dmCpGs associated with the father´s gain or loss in body silhouette, and 12 dmCpGs related to father´s switch in normal or overweight FBS status between voice break and age 30 (Supplementary Data 1A: FBS-V30gr and FBS-V30rs).

The change in FBS DMR includes sites from *NFYA, LARS2, NAP1L5* and *CREBBP* for FBS-V8c, *BLCAP;NNAT, PTCH1* and *GABRG1* for FBS-V8rs, *PM20D, C22orf45, MBP, NUP210L, TRPM4, PACSIN1, ERICH1, MIR886, IQSEC3, BCL11B, AGPAT2* and *NFYA* for FBS-V30c, and *NAPRT1, NUP210L* and *NAPRT* for FBS-V30rs. Details of the DMRs are shown in Supplementary Data 2.

In DMR analysis with DMRCate at FBS-V8c, we identified 13 DMRs including gene loci *FBXO47* (10 sites), *CRISP2* (12), *UPB1;ADORA2A-AS1* (14) which were enriched for many metabolic pathways (see Supplementary Data 2 DMR:FBS-V8c). For FBS-V30c, DMRcate analysis identified 81 DMRs including loci *WT1* (5 sites), *NNAT;BLCAB* (37), *AURKC* (12) and *NUP110L* (10). They were enriched for glycerophospholipid metabolism, vasopressin-regulated water reabsorption, ubiquitin-mediated proteolysis, glycerolipid metabolism and phospholipase D signalling pathway (see Supplementary Data 2 DMR:FBS-V30C_DMRcate).

In the EWAS models investigating change in FBS across adolescence from age 8 to voice break we found overlap with 29 TFs (*q*-value < 0.05) while gain/reduction in FBS from voice break to age 30 overlapped with 33 TFs (*q*-value < 0.05) including *PPARG* (cg22681255 and cg15965578) and *HOXA3* (cg135272218). The list of TFs for each EWAS is provided in Supplementary Data 3.

For a change in FBS from voice break to age 30 (FBS-V30c), out of 791 dmCpGs, 329 showed significant correlation with gene expression (*p* < 0.05) including *VTRNA2-1, TRPM4, GPRC5C, WDR97, ZNF695, NUP210L, GSE1, DIP2C, RP5-894D12.3, FAM26F, C5orf66, AC098614.2, SOCS1, SLC24A4, R3HDM4, ZNF600, RP11-715J22.6, BCL11B* and *MBP*. Of these, 218 dmCpGs were associated with promoter regions (Supplementary Data 4: FBS-V30c). Among the identified dmCpGs in this EWAS analysis, we also observed a strong correlation with gene expression in *ZNF695* (11 sites) and *ZNF600* (7 sites). There were 6 sites each for *CEP85, SOAT2* and *CD52*, and 5 sites for *ZNF334, ZBTB16* and *AGAPT2*, as shown in Fig. 4.

Several of the dmCpGs identified in relation to changes in FBS from age 8 to voice break (*n* = 94) showed correlation with gene expression and

**Table 4 | dmCpGs for father's body silhouette at voice break (FBS-V) FDR ≤ 0.05**

| Name | Effect size | Average methylation | Island | Gene name | Adj. *p*-value |
|---|---|---|---|---|---|
| cg20975419 | 0.09 | 0.18 | OpenSea | | 0.000 |
| cg11789449 | 0.20 | 0.60 | OpenSea | *KCNJ10* | 0.000 |
| cg06444433 | −0.05 | 0.84 | OpenSea | *FERMT1* | 0.015 |
| cg23653826 | 0.03 | 0.93 | OpenSea | *NCK2* | 0.016 |
| cg09655253 | −0.09 | 0.82 | OpenSea | *MOCS1* | 0.016 |
| cg12026976 | −0.01 | 0.92 | OpenSea | | 0.016 |
| cg12587260 | 0.02 | 0.15 | N_Shore | *PAPLN* | 0.016 |
| cg19430728 | 0.01 | 0.94 | OpenSea | *CD44* | 0.020 |
| cg02157155 | 0.09 | 0.54 | OpenSea | *EFCAB9* | 0.022 |
| cg05144772 | −0.02 | 0.90 | OpenSea | *LOC389602* | 0.024 |
| cg25020933 | 0.02 | 0.08 | OpenSea | *B4GALNT4* | 0.024 |
| cg21812470 | 0.06 | 0.20 | N_Shelf | *CNGA1* | 0.026 |
| cg25380281 | 0.03 | 0.91 | OpenSea | *WWP1* | 0.026 |
| cg15549838 | 0.01 | 0.90 | OpenSea | *FNDC7* | 0.026 |
| cg22341132 | 0.04 | 0.44 | Ope nSea | | 0.026 |
| cg13698153 | 0.02 | 0.12 | Island | *VSIG10* | 0.031 |
| cg05839509 | −0.01 | 0.93 | OpenSea | *KIAA0040* | 0.033 |
| cg13242924 | 0.01 | 0.10 | OpenSea | | 0.035 |
| cg04165857 | 0.01 | 0.93 | OpenSea | | 0.035 |
| cg10053674 | −0.02 | 0.89 | OpenSea | *LOC102723544/SLC6A13* | 0.035 |
| cg11737070 | −0.03 | 0.79 | OpenSea | *TBC1D4* | 0.035 |
| cg18092219 | 0.01 | 0.10 | OpenSea | | 0.035 |
| cg11278727 | 0.02 | 0.07 | OpenSea | | 0.036 |
| cg05509659 | 0.00 | 0.02 | Island | *ROBO3* | 0.037 |
| cg06520845 | −0.04 | 0.78 | OpenSea | *EGFR-AS1 EGFR* | 0.040 |
| cg05357152 | 0.03 | 0.23 | Island | *GATA5* | 0.040 |
| cg02212575 | −0.01 | 0.92 | S_Shore | | 0.044 |
| cg04544017 | 0.02 | 0.86 | N_Shelf | *SPATA2L* | 0.047 |
| cg25361524 | −0.02 | 0.83 | OpenSea | | 0.047 |
| cg24052851 | 0.00 | 0.98 | Island | | 0.054 |
| cg16612995 | −0.01 | 0.94 | Island | *ADAMTSL4* | 0.055 |
| cg24420089 | 0.02 | 0.37 | N_Shore | *PTDSS2* | 0.055 |
| cg11949388 | 0.03 | 0.86 | OpenSea | *FOXO3* | 0.055 |
| cg27113059 | 0.00 | 0.02 | Island | *LIPG* | 0.056 |
| cg18109649 | 0.05 | 0.80 | OpenSea | *MRPS28* | 0.056 |
| cg15673187 | 0.01 | 0.04 | Island | *DST* | 0.056 |
| cg15009114 | −0.01 | 0.94 | N_Shore | *RASL11A* | 0.056 |
| cg24906129 | 0.02 | 0.91 | OpenSea | | 0.056 |
| cg07405570 | 0.02 | 0.92 | OpenSea | *LPCAT3* | 0.056 |
| cg12110395 | 0.01 | 0.98 | S_Shore | *ZNF570* | 0.056 |
| cg11720773 | 0.04 | 0.24 | OpenSea | *SYT1* | 0.056 |

multiple sites mapped to genes including *NAP1L5, PTPRN2, CACNA1B, AJAP1, CREBBP, ERGIC1, KCNB1, TBX5, CFLAR, AMH, FBXO47, DIRC3, DLL1, APIP, TIMD4, RP1-34B21.6* and *GALNT14* (Supplementary Data 4: FBS-V8c). The effect of methylation located in a promoter region on gene expression across all tissue types for FBS-V8c is shown in Fig. 4. It shows a strong correlation for genes *NAP1L5* (3 sites), *ATF6B* (3 sites), *RXRG, CDH22* and *LEPROT.*

Enrichment for gene ontology terms: To characterise the biological function of the dmCpGs, each dmCpG was mapped to the nearest gene for each EWAS and detailed functional gene ontology terms for

each gene for each EWAS were retrieved from String. The detailed gene descriptions, related GO terms and lipid-related traits are provided in Supplementary Data 6. The largest GO terms were for genes related to dmCpGs identified in the EWAS on change in FBS from voice break to age 30 (FBS-V30c) $n = 106,616$. In the look-up for 'lipid' related terms, we identified 66 GO terms including lipid metabolic process with 29 associated genes (*ABO, ACSF3, ACSM6, AGPAT2, AGPAT5, AKR7A2, CDIPT, CHKA, COQ2, CPT1A, DHRS11, FABP5, GPIHBP1, HSD11B2, KIT, LMF1, LPCAT4, MBTPS1, NFE2L1, PIGT, PLD3, PRKAG2, SERPINA12, SOAT2, SOCS1, ST3GAL6, SYNJ1,*

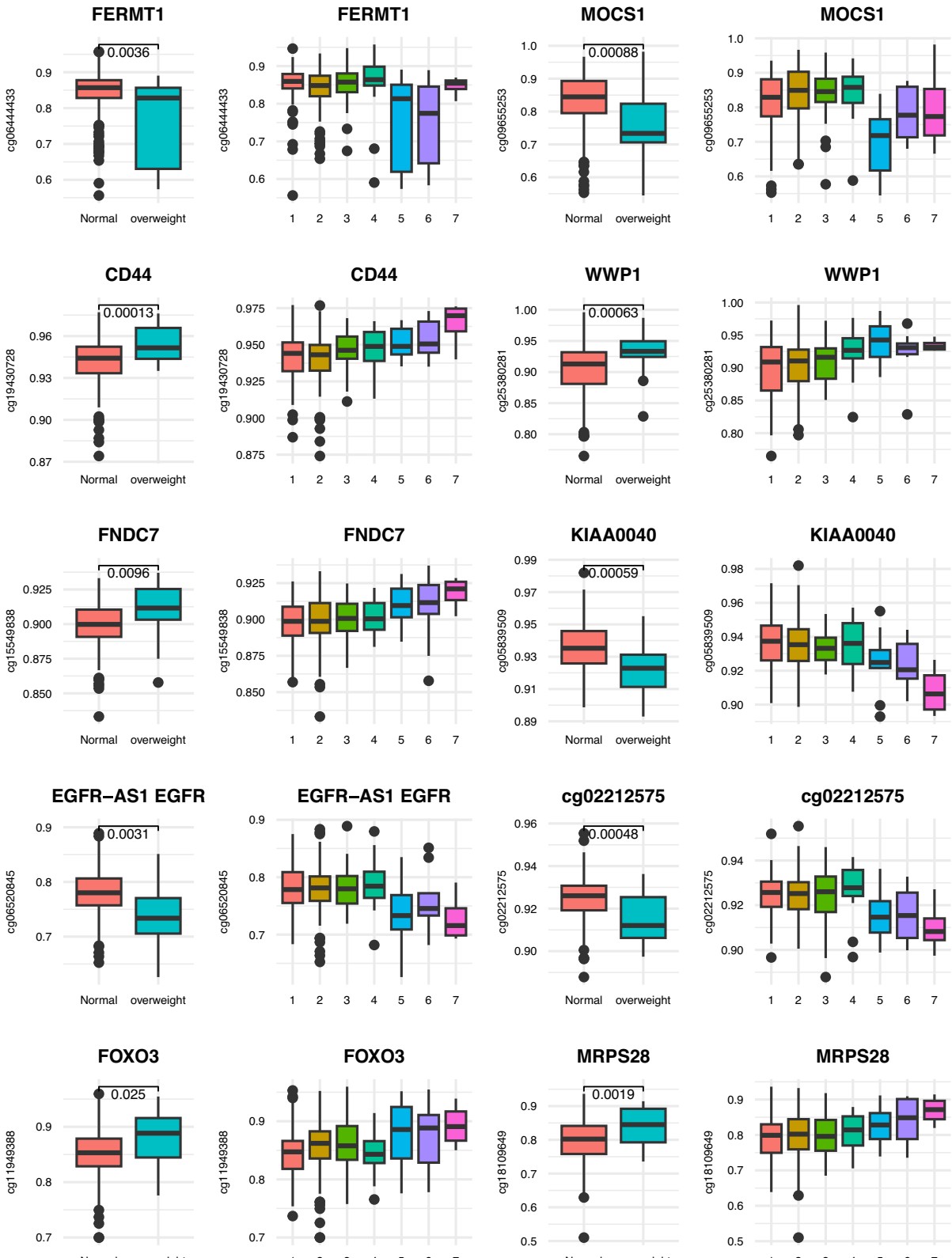

**Fig. 1 | Box plots showing the distribution of methylation levels (beta-values) of top 10 dmCpGs (FDR *p*-value) for FBS at voice break.** The distribution of each dmCpG for the father's "normal" vs. "overweight" body silhouette, and across the father's body silhouette numbers 1–7 is shown. The *p*-value comparing normal vs. overweight is shown above the box plot for each dmCpG.

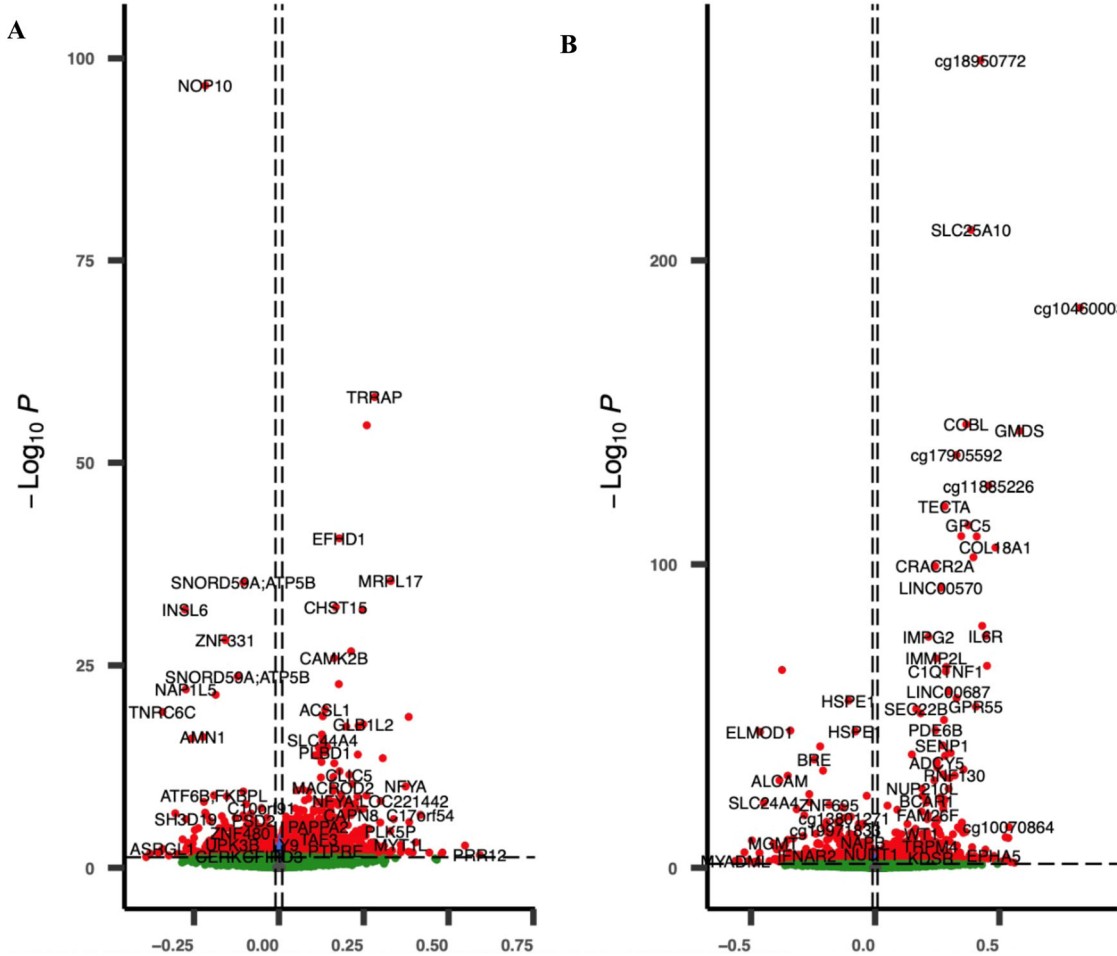

**Fig. 2 | Volcano plot for regression coefficients of dmCpGs associated with a change in father's body silhouettes. A** Between voice break and age 8 years (FBS-V8c) and **B** between voice break and age 30 years (FBS-V30c). The *X*-axis shows regression coefficients, and the *Y*-axis represents −log10 of the *p*-values. Positive coefficients show hypermethylated dmCpGs, while the magnitude shows the strength of the association.

*TPRA1* and *VAC14).* The top associated GO terms and shared genes are shown in Fig. 5.

For dmCpG-related genes in the EWAS on change in FBS from age 8 to voice break (FBS-V8c) there were 97 'lipid' related GO terms, where lipid or lipoprotein measurement was represented by 36 genes *(AGPAT3, ATP8A2, BTNL2, CAMK2B, CAMTA1, CCDC60, CDA, CDC42BPA, CDH13, COL4A2, CYP2B6, DCLK2, DGCR8, DSCAM, FRMD4A, HERC3, LAMC2, LEPR, LRP1, MACROD2, MGMT, MORN1, NAV2, NCOR2, PAPPA2, PHYHIP, PKNOX2, PPP1R13B, PTPRN2, RBMS3, RPA2, SHROOM3, SLC39A11, SLC44A4, SPTBN2, TIMD4).*

For genes linked to dmCpGs associated with a change in FBS from age 8 to voice break (FBS-V8c), the KEGG pathway enrichment included: O-glycan biosynthesis *(GALNT14, XXYLT1)*, Hippo signalling regulation pathways *(CDH13, EPHA2, FGFR2, PRKACA, PRKCH)*, Adipocytokine signalling pathway *(ACSL1, LEPR, RXRG)* and Cholesterol metabolism *(LRP1, LRPAP1)* (see Fig. 6 and Supplementary Data 7 : FBS-V8c).

For dmCpG-related genes associated with a change in FBS from voice break to age 30 (FBS-V30c), the KEGG pathway showed lipid metabolic associated signalling pathways including NRP1-triggered *(AKT3, BCAR1, CDH5, CHD2, COL1A1, EGFR, MAP2K1, RAC1)*, HIF-1 *(AKT3, EGFR, IGF1R, IL6R, MAP2K1, MKNK2, NOS2, NOS3, STAT3)*, Adipocytokine *(AKT3, CAMKK2, CPT1A, PRKAG2, RXRG, STAT3)*, Leptin *(MAP2K1, NOS3, RAC1, STAT3)*, Sterol regulatory element-binding proteins (SREBP) *(MBTPS1, MED15, NFYA, PRKAG2)*, PPAR *(CPT1A, FABP5, RXRG)*, Sphingolipid *(ABCC1, AKT3, GNA12, GNAI1, MAP2K1, NOS3, PPP2R2D,*

*PPP2R5B, RAC1)* and GABAergic synapse *(ADCY5, CACNA1C, GABBR2, GABRP, GNAI1, GNAO1, KCNJ6)*. KEGG pathway network and shared genes are shown in Fig. 6. All identified pathways are shown in Supplementary Data 7.

GO terms: To identify the overlap of dmCpG mapped gene lists with known obesity-related traits in the GWAS catalogue, we used FUMAGWAS (*p* < 0.05) and Enrichr (dbGAP). For FBS at voice break in the female strata, we identified 11 traits and 20 genes related to obesity. These include 2 traits, and 18 obesity-related genes identified in the EWAS on change in FBS from age 8 to voice break (FBS-V8c) and 31 traits in the EWAS of change in FBS from voice break to age 30 (see Supplementary Data 8: FBS-V30c). The latter included waist circumference adjusted for body mass index (*n* = 36), adult body size (*n* = 28), apolipoprotein A1 levels (*n* = 23), waist-to-hip ratio adjusted for BMI (*n* = 34), offspring birth weight (*n* = 11), type-2 diabetes (*n* = 25), appendicular lean mass (*n* = 26) and hip circumference adjusted for BMI (*n* = 31). The top look-up from dbGAP for the EWAS model of difference in FBS from age 8 to voice break (FBS-V8c) was cholesterol and HDL *(RNF157, CHCHD3, BORA, PTPRN2, ERBB4, DEFA3, ROR1, NPHP4, TACC2, CDC42BPA, RXRG, RBMS3)*. For the EWAS on change in FBS from voice break to age 30 (FBS-V30c) the top look-up was BMI *(ATL2, XYLT1, AGAP3, ITGAE, LDLRAD4, ASB18, GNAI1, GPR176, ADAMTS2, IMMP2L, SCML4, MTHFD1L, SCFD2, HAS3, OPCML, NTNG1, KDM4C, SFMBT1, COG5, LMNTD1, PCGF3, KCNIP4, RASSF8, WNT9A, MRPL52, NCAPD3, WDFY4, STAM2, NFE2L1)*. The list of GWAS catalogue enriched obesity-related traits is provided in Supplementary Data 8.

**Fig. 3 | Manhattan plots showing the genome-wide distribution of dmCpGs associated with a change in father's body silhouette. A** Between age of 8 years and voice break (FBS-V8c) and **B** between voice break and age 30 years (FBS-V30c). The *x*-axis shows the position across autosomal chromosomes. The *y*-axis represents −log10 of the *p*-value for each dmCpG (indicated by dots). The green dots show loci with more than 1 dmCpG at FDR-corrected $p < 0.05$. The top dmCpGs on each chromosome were annotated to the closest gene. For intergenic dmCpGs, CpG name was used.

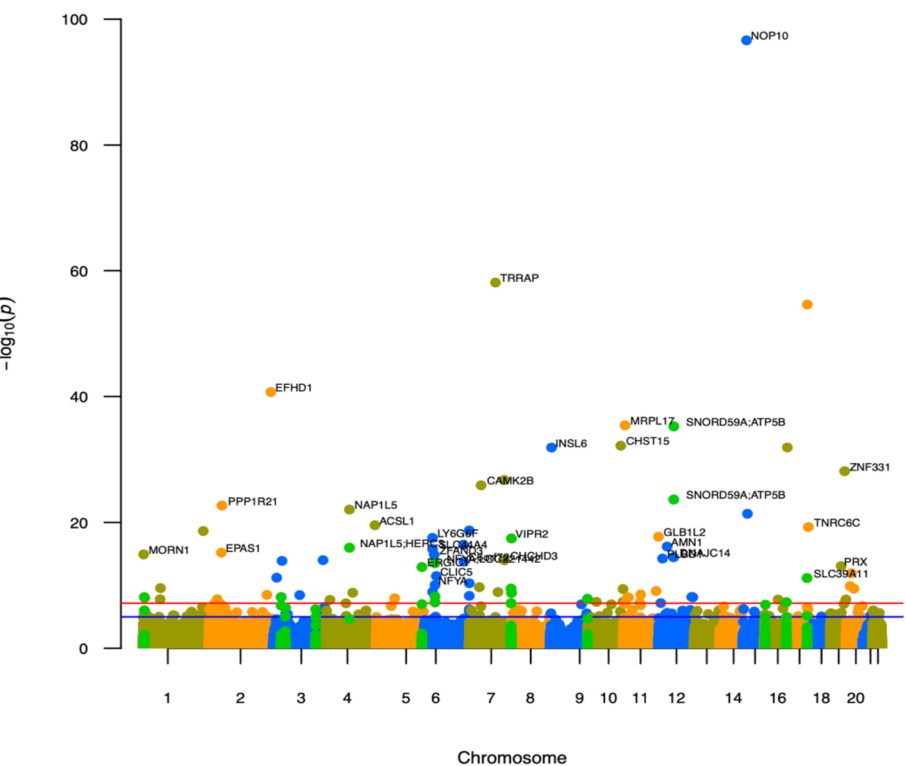

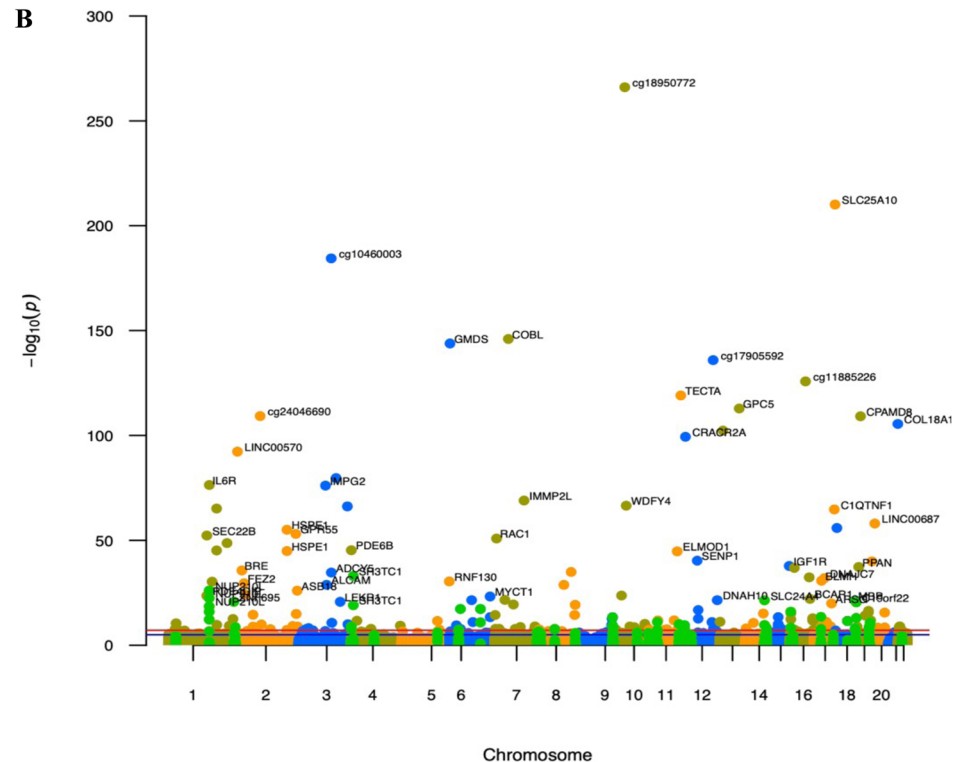

## Biological interpretation of dmCpGs

EWAS atlas lookup: We interrogated the EWAS atlas (7/02/2025) for known trait associations and identified 835 dmCpGs showing 2005 known associations for 217 traits (Supplementary Data 9). The top associated traits include smoking status ($n = 193$), aging ($n = 188$), Down's syndrome ($n = 123$), ancestry ($n = 89$), asthma ($n = 35$) and obesity ($n = 21$).

Correlation between dmCpG methylation and gene expression: For each EWAS, using dmCpG as input, we assessed the known regulatory effect of methylation on gene expression across 6 tissue types (testis, stomach, colon, brain, liver and kidney) using the EWASAtlas (https://ngdc.cncb.ac.cn/ewas/atlas, accessed 7/02/2025). Out of 2005 dmCpG sites, 776 (38%) showed significant correlation with gene expression and we identified 2627

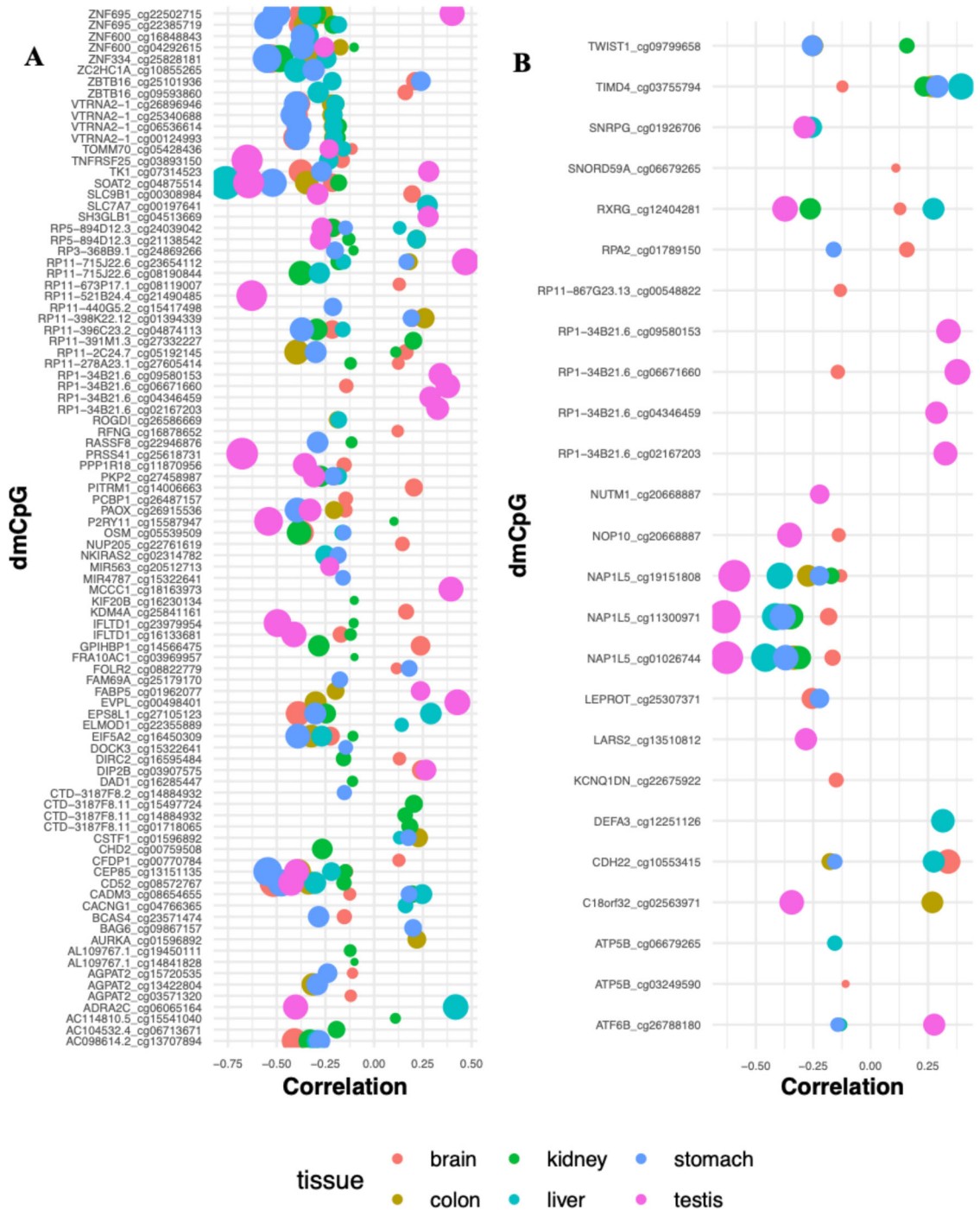

**Fig. 4 | Correlation of methylation level with gene expression for dmCpGs located on gene promoters across 6 tissues, as associated with change in father's body silhouettes. A** Between voice break and age 30 years (FBS-V30c) (4 sites for *VTRNA2-1* and *AGPAT2* and 2 sites for *IFLTD1, ZNF695, ZNF600* and *ZBTB16*) and **B** between voice break and age 8 years (FBS-V8c) (3 sites for *NAP1L5* and *ATP5B*). The dot size corresponds to the level of correlation.

associations between dmCpGs and gene pairs (see Supplementary Data 4 for each EWAS).

The detailed results are provided for overlap with known transcription factor (TF) binding sites, association with other traits (Supplementary results and Supplementary Data 5), KEGG signalling pathway (Fig. 6 and Supplementary Data 7) and obesity-related traits in the GWAS catalogue (Supplementary results and Supplementary Data 8).

Lookup for the effect of genetics on methylation (meQTL): To identify methylation as a consequence of known genetic variants, we searched for dmCpGs using goDMC (designed for 450K $n = 420,509$ CpGs) and the mQTL EPIC database ($n = 724,499$ dmCpGs). We found 698 (35%) and

1209 (60%) dmCpGs that showed meQTL association in the goDMC and meQTL epic database ($p < 0.05$) respectively. Full details of the dmCpGs and associated SNPs are provided in Supplementary Data 10A and B.

Look-up for imprinting and metastable epialleles: To investigate whether the methylation sites associated with father's preconception body silhouettes were related to imprinted genes, we used the Geneimprint database at geneimprint.com (accessed on 17 January 2025). We identified 47 dmCpGs mapped to 24 genes known to be imprinting genes. Some of these genes were represented by more than one dmCpG, including 6 dmCpGs for *BLCAP* and *WT1*, 4 for *VTRNA2-1 (MIR886)*, 3 for *NAP1L5* and *PTPRN2* and 2 each for *MAGI2, MEG3, HOXA2, RASGRF1,*

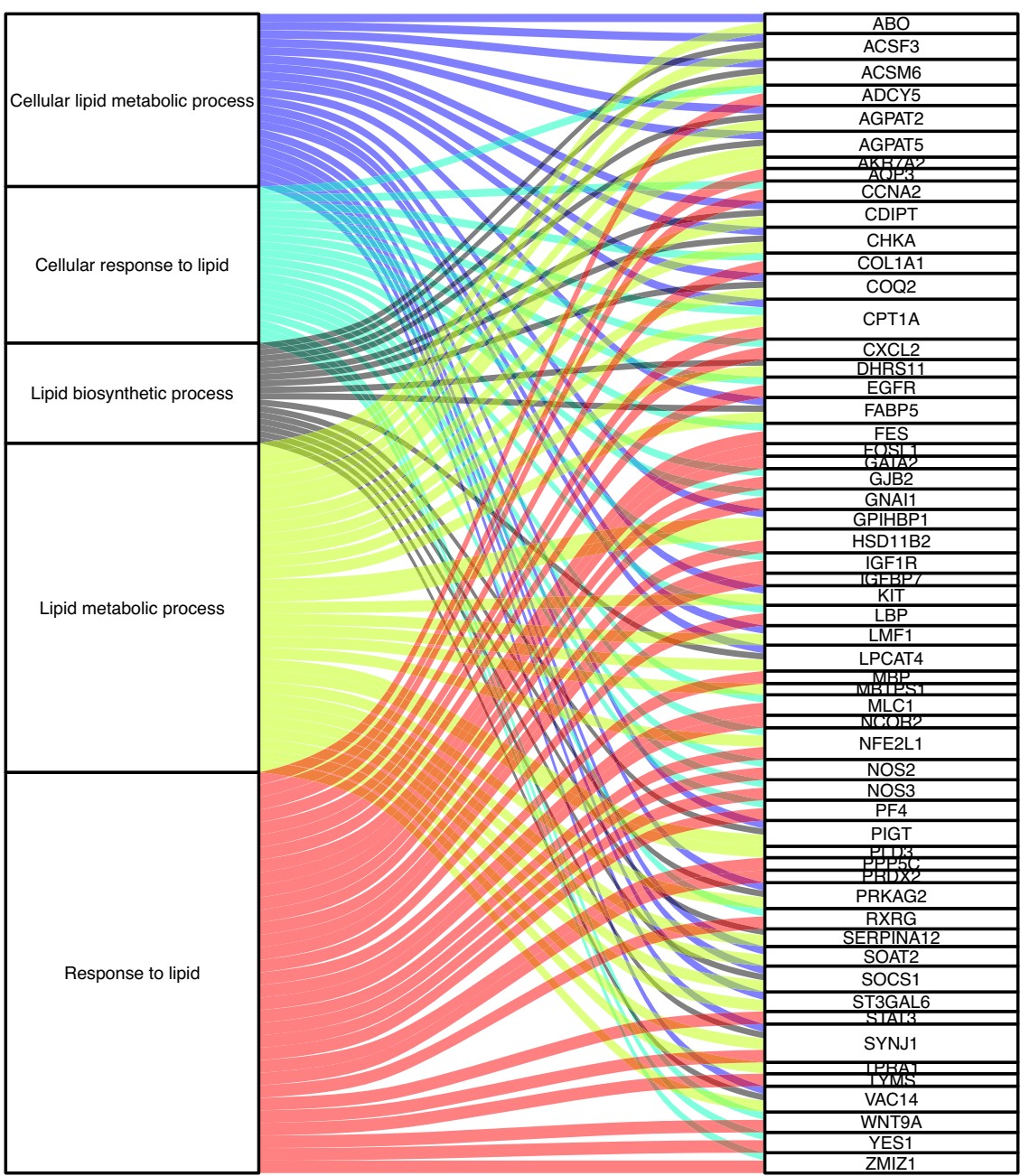

**Fig. 5 | Alluvial plot showing the top 5 lipid-related traits gene ontology terms and gene set for dmCpGs associated with father's body silhouette between voice break and age 30 years (FBS-V30c).** Additional GO terms and related genes are provided in Supplementary Data 6.

*B4GALNT4* and *ZFAT*. Of these genes, 13, 10 and 1 were known to show maternal, paternal and isoform-dependent allelic expression, respectively (Supplementary Data 11).

In the comparison of our unique dmCpGs ($n = 1962$) with known human metastable epialleles ($n = 2408$)[59] we identified an overlap of 37 dmCpGs (Supplementary Data 10: metastable epialleles). In the EWAS Atlas look-up, these dmCpGs showed known associations with paternal uniparental disomy ($n = 10$), aging ($n = 8$) and Down's syndrome ($n = 7$) (Supplementary Data 11: EWASAtlas_metastableEpialelles).

Six dmCpGs were identified by both methods, 2 from *NAPL1L5* and 4 from *VTRNA2-1*; both genes are paternal imprinting genes and metastable alleles (Supplementary Data 10: metastable imprint). In the EWASAtlas *NAPL1L5* (cg01026744) was linked with paternal uniparental disomy. For *VTRNA2*-1, all four dmCpGs (cg00124993, cg06536614, cg25340688 and cg26896946) were linked with Down's syndrome, Parkinson's disease and

breast cancer, while three were linked with gestational diabetes, Clopidogrel resistance, and glycaemic response to glucagon-like peptide-1 analogue therapy in type 2 diabetes mellitus (Supplementary Data 11: ImprintedGene_EWAStraits).

## Association of identified dmCpGs with offspring health outcomes: Asthma, lung function and BMI

Asthma: In total, 119 dmCpGs associated with father´s body silhouette across adolescence showed an association with offspring asthma ($p < 0.05$) (Supplementary Data 12A). The paternal phenotypes showing the largest number of dmCpGs associated with offspring asthma were in the EWAS analysis of change in FBS from voice break to age 30 (FBS-V30c) $n = 52$ (2 sites from *BCL11B*). Other associations with offspring asthma included dmCpGs associated with FBS at voice break in female offspring (FBS-Vf) $n = 19$ and dmCpGs associated with change in FBS from age 8 to voice break

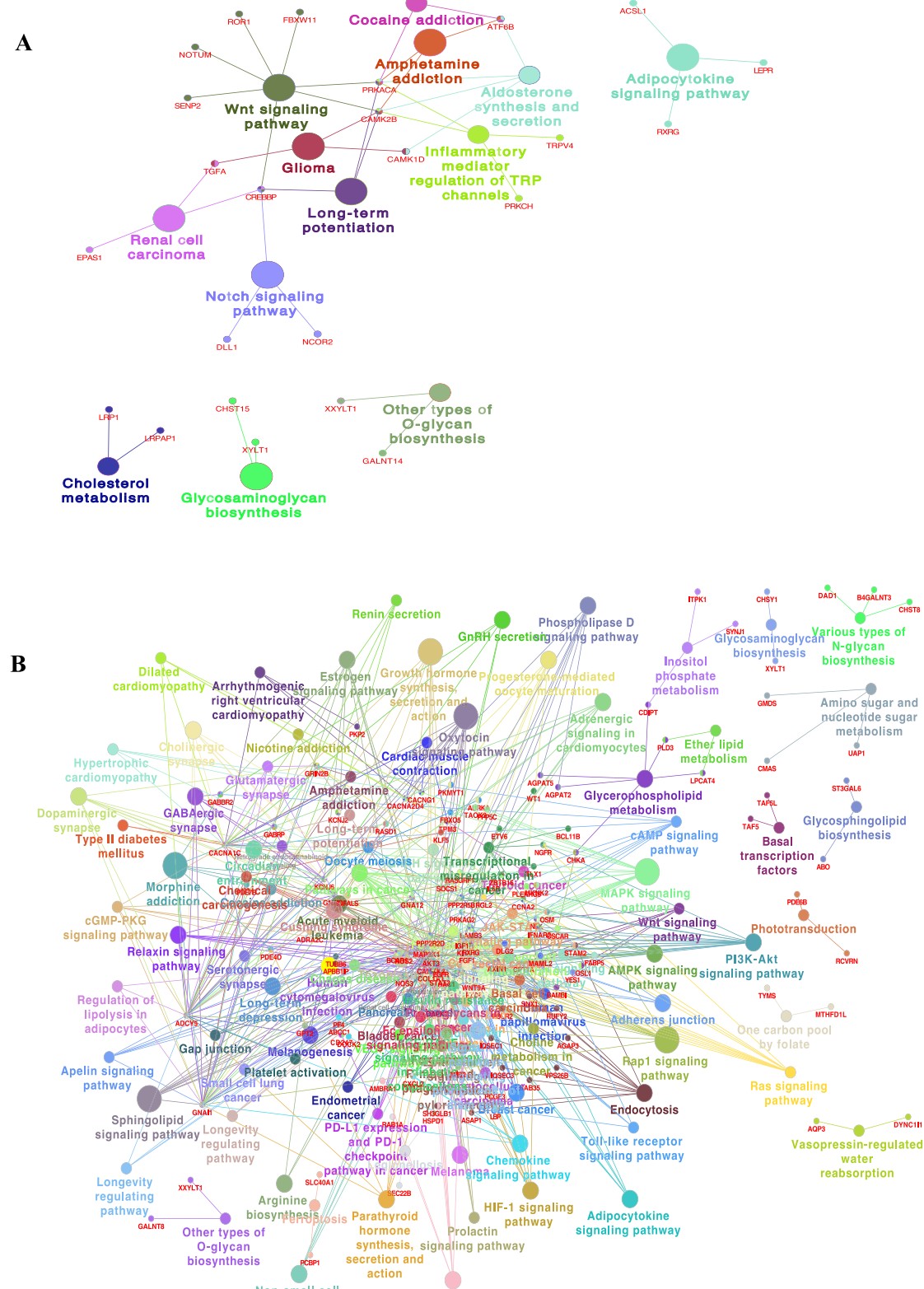

**Fig. 6 | Network plot showing KEGG for dmCpGs. A** Between voice break and age 8 years (FBS-V8c). **B** Between voice break and age 30 years (FBS-V30c). Detailed pathways are included in Supplementary Data 7.

(FBS-V8c) $n = 14$ as well as dmCpGs related to switch in normal or over-weight FBS status from age 8 to voice break (FBS-V8rs) (2 sites from *BLCAP* and *HIST1H2BE*).

Lung function: Analysis of the 2005 dmCpGs in offspring associated with father's body silhouette phenotypes showed that methylation at 982

dmCpGs were associated with different measures of lung function in offspring including pre-bronchodilator FEV1 ($n = 151$), FVC ($n = 145$), and FEV1/FVC ($n = 178$), FVC post ($n = 73$), FEV1 post ($n = 154$) and FEV1/FVC post ($n = 281$). Again, the FBS phenotype dmCpGs showing the greatest number of associations with offspring lung function were

with measures of FBS change across adolescence (Supplementary Data 12B).

BMI: Across all the 2005 dmCpGs identified for all FBS phenotypes, 291 dmCpGs showed an association with offspring BMI ($p < 0.05$). The strongest associations with offspring BMI were seen for dmCpGs associated with a change in FBS between voice break and age 30 (FBS-V30c) where 110 dmCpGs showed nominal association including *DIP2C* (3 sites), *NFYA;-LOC221442* (2 sites) and *IQSEC3* (2 sites). For dmCpGs associated with a change in FBS from age 8 to voice break (FBS-V8c), there were $n = 82$ dmCpGs including *DIP2C* (3 sites), *NFYA;LOC221442* (2 sites) and *IQSEC3* (2 sites) that were nominally associated with offspring BMI. In female offspring dmCpGs associated with FBS at voice break (FBS-Vf), there were $n = 53/370$ (2 sites from *FERMT10*) that were associated with offspring BMI.

When categorising offspring BMI (Normal, Overweight and Obese) 225/2005 dmCpGs showed a nominal association. Detailed information is provided in Supplementary Data 12C.

### Offspring current BMI and current body silhouette

To establish whether offspring DNA methylation associations with fathers' body silhouette phenotypes were confounded by shared environmental exposures, we conducted EWAS of offspring's current BMI and current offspring body silhouette to compare shared dmCpGs between paternal and child exposures. We found 182 dmCpGs associated with offspring BMI and 1 dmCpG associated with offspring body silhouette at FDR corrected $p < 0.05$. The top hit for both was cg20217160 which maps to *LACTB*. When we compared the offspring's top dmCpGs related to current BMI with the top 100 dmCpGs associated with the current offspring's body silhouette, 31 dmCpGs were shared (Supplementary Fig. 2 and Supplementary Data 13). We further compared the regression coefficient between current body silhouette and current BMI using all dmCpGs at nominal $p < 0.05$ ($n = 24217$ dmCpGs). The two regression coefficients showed a correlation $r = 0.97$ with $p < 2.2e-16$ (Supplementary Data 13 and Supplementary Fig. 2), further confirming the validity of the body silhouette approach to measuring offspring adiposity. However, of the 182 dmCpGs identified in the EWAS of offspring BMI only cg19640090 (*ETV6*) of 1962 unique dmCpGs were also associated with FBS phenotypes, suggesting the relationship between offspring DNA methylation and FBS is not significantly confounded by shared familial environment leading to high BMI in both generations.

We conducted an overlap look-up for our 1261 genes (mapped from our 1962 dmCpGs) with obesity-related known genes in open targets (EFO_001073) ($n = 5884$) and DiseGeNET ($N = 2820$) (accessed on 16/07/2022). We identified overlap with 450 genes in either of the two databases while 171 genes were reported by both databases. Our 811 genes (64.3% of 1261) are new reports (see Supplementary Fig. 3 and Supplementary Data 14).

### Discussion

In this study of DNA methylation in humans as related to father's overweight body silhouettes during childhood, voice break, and young adulthood, we identified >2000 differentially methylated CpG sites and many differentially methylated regions that were associated with father's preconception overweight body silhouettes. In particular, changes in the father's body silhouette status across voice break, from normal weight to an overweight FBS or vice versa, were associated with altered DNA methylation patterns in the offspring. This adds to the emerging understanding of the pubertal transition phase to be a period of increased vulnerability for lifestyle influences to drive epigenetic modifications and impact early development and phenotypic outcomes in offspring. This also lends support to our previous study of epigenetic effects in offspring of father's smoking which was by far most pronounced if the father started smoking before age 15 years[9] and by epidemiological studies identifying male prepuberty as a critical exposure window for phenotypic outcomes in offspring[2-7,9].

DmCpGs and DMRs were identified in genes related to insulin-regulation, glucose metabolism, obesity traits, adipogenesis, fat-metabolism, diabetes, asthma, lung function, telomere maintenance, body form and

aging. In agreement with this, a number of identified dmCpGs and DMRs associated with the father's adolescence overweight were also associated with phenotypic outcomes in the offspring such as asthma, lung function and BMI. Our findings thus suggest that epigenetic mechanisms may be important to explaining the associations of father's overweight around voice break with offspring asthma[3,4] height[2] and lung function[2] and, in general, in the transfer of paternal exposure effects to phenotypic changes in offspring.

We have previously established that overweight boys at the age of voice break may impair not only their own health but also the health of their future offspring[2-4]. Supporting a role for DNA methylation driving these phenotype associations, methylation at CpGs in offspring associated with paternal preconception body silhouettes was also associated with offspring BMI, asthma, and lung function. Although, this could potentially reflect a confounding effect of shared familial environment or shared genetic susceptibility between fathers and offspring for these health outcomes, only one site (cg19640090 in *ETV6*) of the 182 dmCpGs identified in the EWAS of offspring BMI was also among the 1962 unique dmCpGs associated with Father's body silhouette phenotypes. This suggests that the relationship between offspring DNA methylation and the father's preconception of overweight body silhouette is not significantly confounded by shared familial environment leading to high BMI in both generations. Similarly, evidence from animal models[8,60] where these confounding factors can be controlled for, suggests a pre-conceptional influence on epigenetic (re) programming events during gametogenesis may, at least in part, be a biological mechanism underpinning these associations.

Examining the dmCpGs in offspring associated with paternal preconception body silhouettes may provide biological insight into the mechanisms linking paternal obesity across puberty with offspring phenotype.

The dmCpGs associated with father's body silhouette at age 8 included cg01945624 located to *SH3TC1* (a gene known to be associated with low lipoprotein), cg10366797 in *HOXC4* (associated with waist-to-hip ratio adjusted for body mass index[61] and body shape index[62]) and cg08975641 mapped to *TH2LCRR* (a gene known to give higher susceptibility to asthma and allergic disease by impact on Th2 cell activity[63]). Genomic loci in *SH3TC1*[15] and *HOXC4*[17] have previously been shown to be differentially methylated in the sperm of obese compared to normal-weight men. Although the sites identified in our study seem to be novel, we suggest this adds support to the conclusion that the methylation signals detected in our study are indeed related to overweight.

The top hit for the father's body silhouette at voice break was cg11789449 in *KCNJ10*, which was hypermethylated in offspring whose fathers had an overweight FBS status at voice break. *KCNJ10* is important in the permeability of pancreatic beta cells which release insulin, and is linked with diabetes[64]. Several other dmCpGs associated with father's FBS status at voice break were also linked to genes with roles in adipogenesis: *LIPG* locus (cg27113059) is known to be associated with lipid traits[65], HDL cholesterol level[66], cardiovascular risk[67] and reduced visceral adiposity[68]; *LPCAT3* (cg07405570) is known to regulate triglyceride secretion[69]; *PTDSS2* (cg24420089) is known to be correlated with fat mass and BMI[70]; *TBC1D4* (cg11737070) regulates insulin-stimulated glucose uptake[71] and is associated with severe obesity, insulin resistance[72] and type 2 diabetes[73,74]; *NCK2* (cg23653826) is involved in regulating adipogenesis[75,76] and DNA methylation sites located within *NCK2* have also been demonstrated to be associated with BMI and body composition in children[77]; *FOXO3* (cg11949388) regulates lipid accumulation and adipocyte inflammation in adipocytes[78]. We also observed that the methylation pattern of cg25380281 in the *WWP1* gene showed an increasing trend across body silhouette scales 1–5 (Fig. 1). *WWP1* regulates adipogenesis and metabolism[79], enhances glucose metabolism[80] and protects against oxidative stress[80]. Interestingly, as for father's body silhouette at age 8, we also observed that several of the annotated genes related to father's body silhouette status at voice break, have previously been reported to be differentially methylated in mature spermatozoa of overweight men, such as those found by Donkin and colleagues[15] (*LIPG*, *PTDSS2*, *NCK2*, *FERMT*, *MOCS1*, *CD44*, *FNDC7*, *SLCGA13*,

*ROBO3, GATA5, SPATA2L, B4GALNT4*) and in a recent study by Keyhan et al.[17] (*B4GALNT4, CNGA1, RASL11A, NCK2, KIAA0040*). Our results add support to the existing literature and to the growing evidence that BMI-related alterations in sperm DNA methylation indeed can be transmitted to the offspring[11,16].

Many observational studies have shown sex-specific differences in disease risk and measurement of obesity traits and BMI[81]. From our sex-stratified EWAS analysis of the father's body silhouette at voice break, we identified different CpG sites for male and female offspring without any overlap. In male offspring, associated dmCpGs were linked to known obesity-related genes including *B4GALNT4* (cg25020933, cg15762098), *ADCY9* (insulin secretion and thyroid hormone synthesis) (cg00610508), *ADRB1* (cg13848598) and *ATP10D* (cg14009629). All these genes have previously been reported to harbour loci that are differentially methylated in obese compared to lean men[15,17]. In a recent study, the imprinted gene *B4GALNT4* has also been associated with adiposity change, both during infancy and in childhood years[82]. In female offspring, out of 308 dmCpGs, 20 genes were known to be related to obesity traits in GWAS catalogues. *FERMT1* (cg06444433, cg16271200 and cg16421850) and *ESRRG* (cg03224209) have putative roles in glucose and fat metabolism. These findings may suggest different underlying molecular mechanisms that can explain the differences between male and female obesity. The DMR analysis in female offspring showed striking signals in obesity-related genes including *RNASE1* (which regulates feeding habit[83] and childhood obesity[84]), *CPT1A* (known to be involved in fatty acid oxidation)[85], *SSTR1* (growth hormone synthesis)[86], *DGKZ* (lipid metabolism)[87] *HIF3A* (childhood obesity)[88] and *ADAMTS16* (related to anorexia nervosa)[89]. Aside from, *SSTR1*, also in the female strata, dmCpG annotated genes have been found in sperm samples of overweight men[15]. *HIF3A* has also been shown to be correlated with paternal BMI and DNA methylation levels in offspring cord blood, although in opposite directions in male and female offspring[16].

Even more pronounced signals of association were identified in the EWAS models of change in father's body silhouette status across sexual maturation. Among dmCpGs associated with a change in FBS between age 8 and voice break (FBS-V8c), 3 sites (cg25607226, cg25890575, cg22991232) mapped to *PTPRN2* which is known to be required for normal accumulation of secretory vesicles in the hippocampus, pituitary, and pancreatic islets. It also plays a role in insulin secretion in response to glucose stimuli and is known to be associated with both childhood[90-93] and adult obesity[15,17]. Four sites were located in *NFYA* (cg09580153, cg04346459, cg02167203, cg06671660) which is related to pre-adipocyte maintenance and/or commitment to adipogenesis[94], energy metabolism, lipid metabolism[95],fatty acid synthesis[95] and leptin gene expression[96]. The top hit, cg20668887 was located in *NOP10* which is involved in ribosomal biosynthesis and telomere maintenance; shorter telomere length is known to be associated with childhood obesity[97], age-related disease[98-100] and BMI[101].

In the EWAS model investigating the switch in normal or overweight FBS status between age 8 and voice break (FBS-V8rs), we identified *PTCH1* which is known to have a role in pulmonary function[102], adult body height[103] and regulating obesity[104]. We also detected 7 dmCpGs located to *GABRG1*, which is a gene known to be linked with childhood obesity, coronary artery disease[105] and adipogenesis[106]. We have reported many dmCpGs that were associated with changes in the father's body silhouette status from voice break to age 30 (FBS-V30c) including cg10460003 in *SLC25A10*, which has a role in fatty acid synthesis[107] and adipocytes insulin sensitivity[108]. Donkin and colleagues also found that sperm DNA methylation patterns in these genes were significantly different in obese men[15].

Six dmCpGs were located in *NUP210L*, which has been shown to be related to asthma and risk of obesity[109] and diabetes[110]. We also identified 6 dmCpGs in *TRPM4* which has a role in vascular formation[111] and is a known target for diabetes[112]. Four sites mapped to *AGPAT2* (cg15720535, cg13422804, cg02703247, cg03571320) which is linked with lipodystrophy[113,114] and has a role in the synthesis of triglycerides and phospholipids[115,116]. The methylation pattern in the dmCpGs sites (cg04292615 and cg16848843) in *ZNF600*, a gene associated with

phospholipid level[117,118] is shown in Fig. 4. Some of the dmCpGs associated with father's switch in normal or overweight FBS status between voice break and age 30 (FBS-V30rs) were also linked to genes known to be associated with regulating adipogenesis (*SERPINA12* (cg01207931), *ZNF423* (cg25096280) and *ADAD2* (cg00225858)). Genomic loci in *ZNF423* have also been shown to be associated with BMI and altered sperm DNA methylation in obese men[15,17].

The DMR genes related to change in FBS are relevant to obesity. *PM20D1* is an enzyme that regulates N-fatty acyl amino acid (NAAs) synthase/hydrolase by regulating the whole body's energy expenditure[119,120]. It converts free fatty acids and free amino acids into NAA. It is a potential anti-obesity target[121] and is associated with cardiovascular risk[119]. *NNAT* regulates metabolic status[122] and has been shown to be differentially methylated in sperm of obese men[15,17] and to be correlated with paternal BMI in offspring cord blood[16].

We found that many of our identified dmCpGs were associated with promoter regions that can influence gene expression, especially in primary organs associated with lipid processing such as the liver. Some of the genes that showed a strong correlation to a reduction in gene expression regulate metabolism in obesity[122], growth restriction and obesity[123], food intake[124] and development of childhood obesity. These include the imprinted genes *VTRNA2-1* (associated with childhood obesity)[125] *NAP1L5* (related to body fat percentage), and *NNAT* (linked to hyperglycaemia and obesity), and the non-imprinted genes *LEPROT* (fat mass), *TIMD4* (total cholesterol measurement), *ATF6B* (phospholipid measurement), and *SOAT2* (LDL, type 1 diabetes, cholesteryl ester measurement, familial lipoprotein lipase deficiency and chylomicron retention disease).

Dysregulation of imprinted genes has been associated with obesity[126], which may also explain why several of the imprinted genes we have identified as related to father's preconception body silhouettes have previously been shown to be differentially-methylated in sperm of overweight/obese compared to normal-weight men. These include *NNAT, RASGRF1, PTPRN2, CDH13, ZFAT, PAOX, CDA, NTM, CHST8, TACC2, DSCAML, C10orf91, JPH3, KCNQ1DN*[15],*PTPRN2, NAP1L5, WT1, HOXA2, B4GALNT, CELF4, DLGAP2*[17,15]), *MEG3* and *NNAT*[16,21,127]. Altered methylation levels in Altered methylation levels in *NNAT* have also been observed in the cord blood of offspring with obese fathers[21], which further supports the idea that epigenetic alterations at imprinted genes in the gametes might be passed onto offspring.

Epigenetic dysregulation of the imprinting genes identified is not only related to obesity but also to diseases including paternal uniparental disomy (*BLCAP* (cg01466133, cg07156273, cg10981598, cg14469070, cg18433380, cg23605670), *NAP1L5*[128] (cg11300971, cg01026744), *MEG3* (cg08698721), *NNAT* (cg11174847) and Down's syndrome (*BLCAP* (cg07156273, cg10981598, cg14469070), *VTRNA2-1* (cg00124993, cg06536614, cg25340688 and cg26896946)) in the EWASAtlas (Supplementary Data 11).

Intriguingly, evidence suggests that *VTRNA2-1* (paternally expressed) is a metastable epiallele, with stable methylation levels shown to be preserved across populations[129]. Thus, the mechanism by which DNA methylation variability in *VTRNA2-1* is inherited essentially appears to be non-genetic[130]. In this study, 4 dmCpGs (cg26896946, cg25340688, cg00124993, cg06536614) mapped to the promotor region of *VTRNA2-1* showed a correlation with reduced gene expression across five tissue types (Fig. 4A). All four have been linked with Down's syndrome, Parkinson's disease, breast cancer and pre- and post-lenalidomide treatment in patients with myelodysplastic syndrome with isolated deletion (5q). Furthermore, gestational diabetes, Clopidogrel resistance[131], and glycaemic response to glucagon-like peptide-1 analogue therapy in type 2 diabetes mellitus are each associated with three of the four dmCpGs in *VTRNA2-1*[132]. It is also known to be associated with BMI and insulin[125] and sensitivity to peri-conceptional environmental exposure[133] and glucose metabolism[134].

For *NAP1L5* (also paternally expressed) 3 dmCpGs (cg19151808, cg01026744, cg11300971) showed correlation with reduction in gene expression. Of these, cg01026744 is a metastable allele linked with paternal uniparental disomy (known to be associated with early-onset of obesity).

Our gene set analysis identified key pathways known to contribute to obesity pathogenesis including: Adipocytokine, *AKT*, *PPAR*, *Wint*, adipogenesis and lipid metabolism.

Of the dmCpGs (genes) associated with offspring health outcomes, *HIST1H2BE* (2 sites) and *TBC1D14* (2 sites) are known to be linked with asthma severity[135], *PHF19* is linked with childhood asthma and *SERPINB9P1* is a known asthma drug target. *NOX3* is also an asthma gene[136] and *SH3TC1* is linked with COPD[137]. Our top hits *KCNJ10* and *FERMT1* also showed association with asthma. In the female EWAS, genes well known to be linked with asthma include *MUC1, RASGRF1* and *IL9*[138]. The lung function-associated loci include dmCpGs from *PTCH1* (cg22073802 and cg16581009), *MUC1, IL9* known to be related to pulmonary function[102] and *SERPINA12*[139]. For dmCpGs associated with offspring BMI, 2 or more sites from *DIP2C, NFYA* and *QSEC3* were identified both at change FBS-V8 and FBS-V30. This suggests that they play a key role in body composition change during childhood and adolescence. Furthermore, voice break-related dmCpGs *FERMT1, NCK2* (adipogenesis), *MOCS1* and *VSIG10* show a clear separation between normal and overweight FBS, suggesting a link with intergenerational BMI.

The main strength of this study is that we have been able to specify the timing of preconception overweight across childhood, voice break and early adulthood in many fathers and have related this to rich data from their offspring including DNA methylation measurements. The current and retrospective body silhouettes have been validated against measured height and weight at different time points in adulthood. Moreover, when we compared the regression coefficients between the offspring's current body silhouette and current BMI in the present study, they showed a correlation $r = 0.97$ with $p < 2.2e{-}16$, which further confirms the validity of the body silhouette approach to measure adiposity. Also, remarkably, many of the annotated genes have previously been linked to obesity and BMI in epigenetic studies on mature spermatozoa[15–17] which adds credibility that the differentially methylated sites identified are truly associated with the father's preconception body composition and overweight, thus represent potential candidates for validation in other studies.

We also acknowledge that our study faces several limitations. The study results are yet to be confirmed in an independent cohort; thus, our findings need further validation. Furthermore, some of the EWAS models assessing timepoints and trajectories of FBS in our study have small sample sizes; despite this, however, significant associations were still identified. Our offspring study population had a large age range, and the subjects come from different study centres and therefore can be considered a heterogenous population. However, we did include age and study centre as covariates in the EWAS regression models to mitigate against these effects and in the RHINESSA cohort. Regarding mothers' overweight status in different time periods and other preconception maternal factors, these were not considered true confounders in the analyses. Previous analyses found these were not associated with adult offspring's asthma[3] and there is minimal overlap in dmCpGs associated with maternal smoking and those associated with paternal BDS (Supplementary Data 9). Potential reverse confounding can however not be excluded.

This study reveals important associations between a father's body silhouette across adolescence and offspring DNA methylation, which strongly supports the idea that a father's preconception metabolic status can impact the epigenome of his future offspring. Further, our study supports the view that the period around puberty may be a particularly susceptible age window for such impact, a concept that may be a game-changer in public health intervention strategies. The identified DNA methylation patterns are related to key signalling pathways known to contribute to obesity pathogenesis and related functions such as insulin regulation, glucose metabolism, adipogenesis, body form, telomere maintenance, asthma and lung function. Our findings showed a 35.7% overlap with previously reported loci linked to obesity, which suggests that about 64% of our genes are novel associations. These sites have the potential to serve as predictive biomarkers for population studies screening for metabolic and respiratory disease, and as therapeutic targets for intervention.

## Data availability

Supplementary Data 1–14 and summary statistics for epigenome-wide association analyses are available from https://doi.org/10.5258/SOTON/D3067. The full data cannot be shared openly in order to protect study participants' privacy, but an anonymised, de-identified version with limited data can be made available on request to allow all results to be reproduced. All requests should be directed to CS, the RHINESSA Study Principal Investigator.

## Code availability

The custom code used to generate graphics are available at GitHub repository: https://github.com/negusse2025/EWAS-of-Father-s-adolescent-body-silhouette-.git.

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

## Acknowledgements

We thank all the study participants, fieldworkers and scientists in RHINESSA, Co-ordination of the RHINESSA study has received funding from the Research Council of Norway (Grants Nos. 274767, 214123, 228174, 230827 and 273838), ERC StG project BRuSH #804199, the European Union's Horizon 2020 research and innovation programme under grant agreement No. 633212 (the ALEC Study), the Bergen Medical Research Foundation, and the Western Norwegian Regional Health Authorities (Grants Nos. 912011, 911892 and 911631). Study centres have further received local funding from the following: Bergen: the above grants for study establishment and co-ordination, and, in addition, World University Network (REF and Sustainability grants), Norwegian Labour Inspection, and the Norwegian Asthma and Allergy Association. Albacete and Huelva: Sociedad Española de Patología Respiratoria (SEPAR) Fondo de Investigación Sanitaria (FIS PS09). Gøteborg, Umeå and Uppsala: the Swedish Heart and Lung Foundation, the Swedish Asthma and Allergy Association. Reykjavik: Iceland University. Melbourne: National Health and Medical Research Council (NHMRC) of Australia (research grants 299901 and 1021275). Tartu: The Estonian Research Council (Grant No. PUT562). Århus: The Danish Wood Foundation (Grant No. 444508795), the Danish Working Environment Authority (Grant No. 20150067134), Aarhus University (Ph.D. scholarship). The body silhouette image used in the graphic abstract and Supplementary Fig. 1 was generated using Clip Studio Paint and kindly provided by Alejandro Villén Real (alejandrovillen@gmail.com).

## Author contributions

C.S., T.M.Ø., M.L., S.A., F.G.R., N.T.K. and J.W.H. contributed to conceptualisation. T.M.Ø., M.L., S.A., N.T.K. and J.W.H. performed data curation. N.T.K., C.S. and J.W.H. carried out formal analysis. N.T.K., S.A., C.S. and J.W.H. provided methodology. C.S. performed project administration. N.T.K., T.M.Ø., M.L., F.G.R., C.S. and J.W.H. performed writing—original draft. N.T.K., T.M.Ø., M.L., S.A., F.G.R., A.M., A.O., B.B., F.J.C.G., L.P.G., M.H., N.O.J., S.C.D., S.M.S., V.S., C.S. and J.W.H. contributed to writing—review, editing and final approval.

## Funding

## Competing interests
The authors declare no competing interests.

## Additional information

[1]Human Development and Health, Faculty of Medicine, University of Southampton, Southampton, UK. [2]Department of Global Public Health and Primary Care, Centre for International Health, University of Bergen, Bergen, Norway. [3]Department of Health and Caring Sciences, Western Norway University of Applied Sciences, Bergen, Norway. [4]Unit of Epidemiology and Medical Statistics, Department of Diagnostics and Public Health, University of Verona, Verona, Italy. [5]Department of Clinical Science, University of Bergen, Bergen, Norway. [6]Department of Medical Sciences: Clinical Physiology, Uppsala University, Uppsala, Sweden. [7]Section of Sustainable Health, Department of Public Health and Clinical Medicine, Umeå University, Umeå, Sweden. [8]Department of Allergy, Respiratory Medicine and Sleep, Landspitali University Hospital, Reykjavik, Iceland Faculty of Medicine, University of Iceland, Landspitali, Iceland. [9]Department of Pulmonology, Albacete University Hospital Complex, Albacete, Spain. [10]El Torrejón Health Centre, Andalusian Health Service, Huelva, Spain. [11]Occupational and Environmental Medicine, School of Public Health and Community Medicine, Institute of Medicine, Sahlgrenska Academy, University of Gothenburg, Gothenburg, Sweden. [12]Centre for Epidemiology and Biostatistics, Melbourne School of Population and Global Health, University of Melbourne, Melbourne, Australia. [13]Department of Public Health, Research Unit for Environment, Work and Health, Danish Ramazzini Centre, Aarhus University Denmark, Aarhus, Denmark. [14]Department of Occupational Medicine, Haukeland University Hospital, Bergen, Norway. [15]NIHR Southampton Biomedical Research Centre, University Hospitals Southampton, Southampton, UK. [16]These authors contributed equally: Cecilie Svanes, John W. Holloway. ✉e-mail: cecilie.svanes@helse-bergen.no

