## [Transparent Peer Review file · Communications Biology]

Father's adolescent body silhouette is associated with offspring asthma, lung function and BMI through DNA methylation

Corresponding Author: Professor Cecilie Svanes

This manuscript has been previously reviewed at another journal. This document only contains information relating to versions considered at Communications Biology.

Version 0:

Reviewer comments:

Reviewer #2

(Remarks to the Author)

The manuscript by Kitaba et al. studied DNA methylation changes in offspring of overweight fathers to determine the effect of paternal body silhouette on DNA methylation signatures of their children. Overall, the study is well designed and the findings will be of interest for researchers working to delineate paternal effects on the offspring's epigenome. However, the authors tried to expand the results section and this made it difficult to read through the manuscript since similar analysis is repeated across different groups. Please try to present your results differently to make it easier for the readers. For example, the gene ontology and KEGG pathway analysis can be directly presented after describing DMPs/DMRs for each group. I also have a few suggestions/comments to help improve the manuscript:

1. The authors keep using logFC to describe methylation differences across their manuscript. First, please indicate that this is log₂FC to avoid any confusion. Second, I do not recommend using log₂FC for DNA methylation differences and recommend the authors to mention the Beta methylation difference instead. The reason for this is because small methylation differences in certain scenarios (for example cases = 1% vs controls 3% average methylation) would have a higher absolute log₂FC when compared to drastic methylation differences for other sites/regions (for example cases = 50% vs controls 90% average methylation). Similarly, in Table 3 the authors mention Average Expression in column 3. This has to be methylation Beta value instead.
2. The GO and KEGG pathway enrichment analysis has to be performed using tools that account for probe bias distribution across genes since EPIC arrays were used. One tool that can be used by the authors is the methylglm function from the methylGSA package.
3. Did the authors only use Beta values in their analysis or were M-values used? M-values are more statistically valid for differential DNA methylation analysis (Please refer to PMID: 21118553).
4. Did the author consider adjusting for any maternal factors that might affect the offspring's epigenome particularly since they focused on paternal effects?
5. Apart from differentially methylated probes (DMPs), I would also recommend that the authors perform a region based analysis focusing on promoters. One would expect an effect of DNA methylation changes on gene expression when it occurs in promoters. Mapping single DMPs back to promoters is not sufficient because it is the average of DNA methylation levels at several CpG sites rather than a single CpG site that would have an effect.

Reviewer #3

(Remarks to the Author)

General

This study is about a timely and interesting topic. Although it is observational, the field needs more similar studies, so data/results could be collected, compared, merged, etc... to find common sense. Hence, it may help researchers moving

forward.

The total study population is not large, but acceptable seen its methodological approach, using EWAS/EPIC analyses. However, some major issues about the subgroups, distribution, potential heterogeneity, clarity of the presentation,... are added below. Methylation outcomes included a comprehensive biological interpretation of dmCpGs (GO, Enrichment, correlation with gene expression, ...). Additionally, the manuscript includes some nicely illustrated figures.

Major comments

The title "Father's overweight in adolescence and offspring DNA methylation" is too brief, and needs more information about the aims or findings (while keeping it scientifically sound).

As the study is about potential outcomes such as asthma, lung function and BMI, at least this should be clear from the title. Next, "overweight" was not measured as such (no BMI in fathers), but self-reported "body shape" (or silhouette was used). Hence, use "perception of body shape" or similar wording...

The abstract starts with a strong statement that has not sufficiently been proven yet, especially in humans. "Boys' prepubertal overweight appears to cause asthma and low lung function in future offspring,"... Causality has not been proven in human studies so far, hence the wording "cause" needs to be replaced, using "associated with", "related to",...

Further, the abstract misses important information, such as the specific method used to identify differentially methylated cytosine-phosphate-guanine (dmCpG) sites, the fact that "body silhouette" was self-reported or a perception (or body image) ... , and the tissue type used to measure dmCpGs needs to be added as well.

Abstract shows 2 short lists of genes, but it is not clear where these came from in the text body ("top hit" is mentioned, but it is not clear what is the meaning of "top hit").

Inter- and transgenerational is not the same; hence, why concluding about "multiple generations"?

Introduction.

Lines 76-77: more information about studies (ref 2-8...) is needed. Are these independent studies? Other than the current one? Performed by other researchers? Association studies?...

It seems that the rationale is based on their own data and the same data set (RHINESSA) as the current one. Hence, it looks like most of this section (rationale and hypothesis) is built on data from the same research group.

However, the literature does show several related reports!

In short, the introduction is too brief and misses "supporting" references about father-child inheritance through epigenetic mechanisms, from other groups.

At least the following (pioneering) references are missing:

DOI: 10.1038/ijo.2013.193 (obese fathers and offspring DNA methylation)
<https://doi.org/10.1371/journal.pone.0218615> (obese fathers and offspring DNA methylation cord blood)
DOI: 10.1016/j.cmet.2015.11.004 (obese men and DNA methylation of sperm)
doi.org/10.1186/s13148-020-00997-0 (male obesity and DNA methylation sperm)
DOI: 10.1186/1741-7015-11-29 (obese men and offspring DNA methylation)
DOI: 10.1038/sj.ejhg.5200859 (intergenerational inheritance...)
DOI: 10.1038/s41586-024-07472-3 (other epigenetic mechanism of inheritance)
<https://doi.org/10.1093/eep/dvy007> (this inheritance even has a "name": POHaD)

About mechanisms and inheritance:

DOI 10.1002/bies.201300113 (father-child and susceptible windows within the sperm epigenome)
doi: 10.1016/j.nut.2003.09.011 (or similar, referring to mechanisms of inheritance)
doi: 10.1001/jamanetworkopen.2019.16777

The introduction also misses more profound insights about epigenetic mechanisms (or their hypothesis about these).

It is not clear how timing of "voice break" relates to "spermatogenesis" and DNA (re)establishment of methylation marks in sperm? Please clarify.

Why do the authors believe that this question is important? Is there a reason why men would remember their weight/silhouette better at the time of voice break?

Method and study (population)

Why beta-values and not M-values? Beta-values do not have a normal distribution and they are heterogeneous, hence do not meet the assumptions of a linear regression.

The manuscript misses the exploration of specific mechanisms of epigenetic inheritance (such as through imprinting). The following should be verified through a focussed additional test: association between FBS and father's DNA methylation at imprinted genes (using data bases such as the Geneimprint data base or the human Imprintome. See:

DOI: 10.1080/15592294.2022.2091815

And/or

<http://geneimprint.com/site/genes-by-species.Homo+sapiens>.

Study population and cofactors:

Age range is large (7 -51 years) for a sample size of about 300 offspring.
Age is a major confounder. Did the authors include findings from other studies to correct for age?
Same issue about smoking status of offspring?

Another important potential confounder that needs to be verified is study centre.
Line 191-192, etc: How did the authors account for the fact that the study has very imbalanced classes (307 vs 32)?
Only 10% of fathers had overweight. Hence, this study is based on only 32 subjects that have been "exposure"; taking into account the large age range and heterogeneity of several other (unmeasured) factors, is a major weakness of this study.
Age range is large (7-51y) and data come from a very heterogenous population, from Europe till Australia.
Six different study centers were included, how was the distribution of overweight, age, ...by centre? It needs an additional demographics table.

Other weaknesses are the use of "silhouettes" using only 2 categories (in the major analysis?). FBS was divided in 2 categories, and from 1 till 7. But no further explanation is given (how were these defined? Calculated?). If published earlier, it could be included in the general figure /overview or as suppl. information.
It is a self-reported "estimate of BMI". The authors refer to validation tests, but these should be added (or repeated) in suppl. data for clarity. References from other studies (performed others) are needed showing that this method is valuable.
There are certainly papers on this, here is one example on reliability of a BMI-based Silhouette Matching Test:
DOI:10.5993/AJHB.27.4.7

Also asthma was self reported, based on one single question: "Have you ever had asthma diagnosed by a doctor?". It was not verified in a medical record afterwards?

It would be informative to add a table on the frequencies of "asthma", "lung function", and frequencies of BMI (or obesity status) in offspring, by centre, by gender, ...

Was there a significant association between socio-economic status of the grandparents and the father's body measurements? It could be added in results or supplementary material how father's measurements differ between socio-economic status of grandparents.

In general, the number of covariates is limited. It is possible that the results are based on unmeasured factors.

Results/methods

The presentation by each "exposure to silhouette of the father by his age" separately, is not sufficient to draw clear conclusions. Because the outcomes of DNA methylation were measured once in offspring (and hence, these methylation outcomes are the same for all exposures through the father).
It is warranted to perform a comparative data analysis using all paternal information in the same model. E.g., using matrices, or using a categorization of windows of exposure, in one mathematical model (e.g., a positive change in silhouette between age 8 and voice break, versus, a positive change after voice break, one single exposure of "overweight", being twice "overweight" in 2 windows, etc...)?
If this could be done/shown, only then this sentence makes sense: Line 888: "Importantly, the period around voice break stands out as the most critical age window regarding epigenetic effects..."

The direction and importance of changes at CpGs have not been elaborated or discussed. Such as increased versus decreased, imprinted (hyper/hypomethylation),...

Only 2 imprinted genes were identified?
Why is this not mentioned in the abstract? As it is important to explain inheritance.
B4GALNT4 in males: In table 3, explain why this CpG has 8 % methylation? (is this imprinted at the paternal allele?)
An interpretation is missing. For instance, this means that if the father had overweight at age of voice break, his son is at risk to have +2% at this site...
Similar comment about FERMT1: 84% methylated in blood.
Note, "AveExpr" is confusing (see Table3), although it may be the output of the program, I assume it is "average methylation" (0 – 1; where 1 is 100%)?

Overall I miss interpretation of the data and explanation of their findings.

Just to give some more examples:

If imprinted, how can methylation marks already established before the age of 11, become aberrantly differentially methylated after prepuberty (at end of puberty/"voice break")?

How can a small effect in sperm (small % of sperm cells) of 32 fathers result in a measurable change in DNA methylation of blood cells in a small number of children? (32 from overweight fathers; e.g., 16 were boys and 16 were girls)?

Lines 464: sensitivity test results need to be added in suppl. tables.

Discussion:

This is overstated and not correct:

(line 358):“In this first study of DNA methylation in humans as related to father’s overweight/obesity during stages of sexual maturation,...”

and

line 456: “to our knowledge, this is the first EWAS to use father body silhouettes in humans to identify obesity-related markers”...

line 460 “that there are no other cohorts with detailed data on overweight during fathers’ preconception lifespan, that allow replication of our findings”...

See other literature where obesity (etc.) in puberty and other periods of timing during spermatogenesis and development of sperm have been studied.

Calling “voice break” (instead of “puberty”), or “silhouette” (instead of “overweight”, in other studies), does not make the current study more unique than earlier studies. Instead, the current study includes self-reported data... while earlier studies included measured data (BMI ...).

Although this study is not unique, consistency in the results is important to show causality. Hence, as stated in their first sentence (“cause of”), there is a need to do comparative analyses with earlier similar publications.

Line 457: “about 70% of the genes identified by the body silhouette related dmCpGs signify novel associations with obesity, not previously captured using BMI measurements.”

Please clearly refer to the table where this (70%) is presented.

Line 893: “smoking before age 15 years”, reference 1 was not the first showing this association.

Please add the paper of Pembrey’s study. doi: 10.1038/ejhg.2014.31.

A general remark: just as in the introduction section, the discussion misses a thorough comparison of these findings with findings published by others on similar exposures and outcomes. As said, published data bases could be used to verify potential overlap or similarity in the results.

Minor comments

Line 76-77 or in line 92: add average known/expected age (or range) of “voice break”.

Is the format of the reference list correctly implemented?

Some papers are indicated as 1 author et al., others are fully mentioned (including all coauthors).

Check and review figure 1 and Supplementary Figure 1 for their scale, clarity, references, scale measures of gradation in body size changes for defining changes in FBS.

Line 157-158: does the FDR are multiple testing adjustment refers to the covariates? It is not clear.

Line 161 etc: Not clear, it seems like offspring was measured at multiple time points...?

Results. Father’s FBS at voice break. Line 213 (same line 225 and throughout the text: “top-hit”): “top-10 dmCpGs” please clarify (based on p-value, magnitude, both, ...?)

Results. Change in FBS between 8 and voice break. Line 231: explain $\lambda=1.05$? (is not mentioned in methods section).

Results. DMRs analysis. Why are the number of differentially methylated regions much lower than CpGs; explain in the discussion.

Results. Offspring’s current BMI and current body silhouette. Line 345: “We found 182 dmCpGs ... and 1 dmCpG associated with offspring body silhouette”, in suppl. Figure 1 shows they are around 100 with 33 overlapping the one of offspring BMI. In general, numbers do not correspond between the figure and text. Where is supplementary figure 2? Line 346: “the top hit for both...” Both which?

Discussion. Line 388-389: “Many observational studies obesity traits and BMI”, reference? Line 410: “shorter telomere length is known to be associated with childhood obesity and BMI.”, ref? Line 419-420: “Six dmCpGs were located to NUP210L ...”, ref? In general, when it is reported “it has been shown, it is known, ..” a reference is needed.

Results. Line 198: “strongest effect”? Larger?

Results. Line 248: misleading title, it seems DMR are found in the father. Maybe specify “offspring DMR”.

Results. Enrichment for gene ontology terms. Line 294: “Largest GO terms” or the one that appears the most?

Figures: Figure 2: Clarify why sometimes CpG sites names are added and some others gene names are added. Also, A and B labels are missing in the graph, so either add it or change in the legend to the right and left graphs.

Figure 3: Clarify why sometimes CpG site names are added and some other gene names are added.

Figure 4: Add a definition of the size of the dots to be more precise, either as a legend in the graph or in the legend of the figure as text.

Figure 6: In the A graph no genes names appear in the graph, whereas they do in the B graph.

Supplementary discussion. Line 982-986: change font.

Suppl. Figure: not clear. Offspring body silhouette and offspring BMI are outcomes, while EWAS 2005 is? This is not explained in the legend.

Version 1:

Reviewer comments:

Reviewer #2

(Remarks to the Author)

Thank you for addressing all my comments.

Point-by-point response to the referees' comments

Reviewer #2

The manuscript by Kitaba et al. studied DNA methylation changes in offspring of overweight fathers to determine the effect of paternal body silhouette on DNA methylation signatures of their children. Overall, the study is well designed, and the findings will be of interest for researchers working to delineate paternal effects on the offspring's epigenome.

We wish to thank the reviewers for their careful review of the manuscript and their interest in the study.

1. However, the authors tried to expand the results section, and this made it difficult to read through the manuscript since similar analysis is repeated across different groups. Please try to present your results differently to make it easier for the readers. For example, the gene ontology and KEGG pathway analysis can be directly presented after describing DMPs/DMRs for each group. I also have a few suggestions/comments to help improve the manuscript:

We agree with the reviewer that the results section could have been presented more clearly.

We have completely rewritten this focussing mainly on voice break and change across voice break in the main text of the paper, as we have found strong methylation signals in these EWAS. We have also used the combined list of dmCpGs associated across phenotypes (N=2005, 1962 unique dmCpGs) for dmCpG interpretation, including known trait overlap and enrichment of imprinting genes. As suggested, we have discussed DMP and DMR and related pathway-enrichment after each EWAS. For transparency, we have also presented results of downstream analysis for each phenotype separately in the supplementary materials.

2. The authors keep using logFC to describe methylation differences across their manuscript. First, please indicate that this is log₂FC to avoid any confusion. Second, I do not recommend using log₂FC for DNA methylation differences and recommend the authors to mention the Beta methylation difference instead. The reason for this is because small methylation differences in certain scenarios (for example cases = 1% vs controls 3 % average methylation) would have a higher abs. log₂FC when compared to drastic

methylation differences for other sites/regions (for example cases = 50% vs controls 90 % average methylation). Similarly, in Table 3 the authors mention Average Expression in column 3. This has to be methylation Beta value instead.

Thank you for this suggestion. Throughout the manuscript we have replaced log2FC with effect size and Average Expression with average methylation (Beta value) as suggested to improve interpretability for the reader.

3. The GO and KEGG pathway enrichment analysis has to be performed using tools that account for probe bias distribution across genes since EPIC arrays were used. One tool that can be used by the authors is the methylglm function from the methylGSA package. Thank you for the suggestion. In the revised version we have regenerated GO pathway enrichment using methylGSA and have provided these as supplementary results alongside the String and KEGG pathways (supplementary Data 5).
4. Did the authors only use Beta values in their analysis or were M-values used? M-values are more statistically valid for differential DNA methylation analysis (Please refer to PMID: 21118553).

We acknowledge that the use of beta or M-values in EWAS is a matter of some debate within the population epigenomics community. A number of studies have used transformations of beta-values, such as using log ratios of methylation percentage (M-values) in order to obtain a normal distribution or regression based on an alternative distribution (e.g. beta regression). However Mansell et al. (PMID 31088362) investigated whether the assumptions of linear regression are satisfied when measuring DNAm as beta-values and whether any violations bias the results of DNAm studies. Their conclusion was that the use of linear regression with beta values in DNAm studies, even if the data do not satisfy the standard assumptions of this test, does not appear to lead to biased results. In their analysis ~70% of DNAm sites on the EPIC array demonstrated significant bias of at least one assumption when using M-values compared to ~71% using beta values. As for beta values, CpG sites at the extreme ends of the DNAm distribution for M-Values were more likely to fail the statistical tests.

For reproducibility, we have also conducted EWAS at voice break using M-values which showed all our dmCpGs were significant at FDR <0.05. Furthermore, we compared the reproducibility (as sample heterogeneity check) by running EWAS with Bergen centre only.

The paired t-test of regression coefficients between the whole cohort and Bergen was not significantly different ($p=0.14$). The correlation between regression coefficient between the cohort and Bergen EWAS was 0.96 at $p<0.05$. In our comparison, 13 dmCpGs showed association at $FDR < 0.05$ in EWAS. This demonstrates that our dmCpGs are reproducible, both in effect size and direction of association (see supplementary Data 1B).

5. Did the authors consider adjusting for any maternal factors that might affect the offspring's epigenome particularly since they focused on paternal effects?

We did not adjust for maternal factors since it can be discussed whether maternal factors preconception, and even in mother's childhood/adolescence, fulfil the criteria for true confounders in our analysis. Further, in our previous work we have shown that in the RHINESSA/ RHINE cohorts, mothers' overweight or smoking status in different preconception time periods were not associated with adult offspring's asthma (Johannessen A, et al. Being overweight in childhood, puberty, or early adulthood: Changing asthma risk in the next generation? *J Allergy Clin Immunol.* 2020;145(3):791-799.e4. PMID: 31505189; Svanes, C et al. Father's environment before conception and asthma risk in his children. *Int J Epidemiol.* 2017 Feb 1;46(1):235-245. PMID: 27565179). In addition, in supplementary table 9 we present the result of EWAS for maternal smoking which shows very little overlap with the paternal obesity associated dnCpGs (55/2002). We have still acknowledged this as a limitation in the discussion (line 679-83).

6. Apart of differentially methylated probes (DMPs), I would also recommend that the authors perform a region-based analysis focusing on promoters. One would expect an effect of DNA methylation changes on gene expression when it occurs in promoters. Mapping single DMPs back to promoters is not sufficient because it is the average of DNA methylation levels at several CpG sites rather than a single CpG site that would have an effect.

We would like to thank the reviewer for these suggestions. In the revised manuscript we have conducted regional enrichment using *goregion* function from *missMethyl* R package for gene enrichment (see supplementary Data 2) and see line 308 - 314.

Reviewer #3

7. General

This study is about a timewise and interesting topic. Although it is observational, the field needs more similar studies, so data/results could be collected, compared, merged, etc... to find common sense. Hence, it may help researchers moving forward.

The total study population is not large, but acceptable seen its methodological approach, using EWAS/EPIC analyses. However, some major issues about the subgroups, distribution, potential heterogeneity, clarity of the presentation, are added below.

Methylation outcomes included a comprehensive biological interpretation of dmCpGs (GO, Enrichment, correlation with gene expression, ...). Additionally, the manuscript includes some nicely illustrated figures.

We appreciate the reviewer's interest in our work. In the discussion we have acknowledged the limitation of sample size (page 22, Line 673-75). However, the minimum sample size per group in the analyses is 32. We also acknowledge the potential heterogeneity in the sample set, but all subjects were of European ancestry and were adjusted for study centre in the analyses. In addition, as a sensitivity analysis at voice break we analysed the Bergen sample alone (the largest study centre, N= 201 (FBS Normal=201, Overweight =19) all our dmCpGs remained significant (see supplementary data 1B) and see response to reviewer 2 comment #4.

In our QC steps to detect technical variability, we used single value decomposition SVD plot from CHAMP package as shown in the figure below. To remove detected variability, we applied batch effect correction using combat on sample batch and slide. As shown in the figure, this reduced the variability in PC1.

We have also re-written the manuscript extensively to improve the clarity of presentation.

Major comments

8. The title "Father's overweight in adolescence and offspring DNA methylation" is too brief and needs more information about the aims or findings (while keeping it scientifically sound). As the study is about potential outcomes such as asthma, lung function and BMI, at least this should be clear from the title.

We agree with the reviewer and have revised the manuscript title to be more informative and include findings and outcomes investigated. The revised title is:

'Father's adolescent body silhouette associated with offspring asthma, lung function and BMI through DNA methylation'

9. Next, "overweight" was not measured as such (no BMI in fathers), but self-reported "body shape" (or silhouette was used). Hence, use "perception of body shape" or similar wording...

We do acknowledge that we used self-reported figural drawing scales to assess body size, and that BMI was not measured at all time points in the fathers. In the revised version of the manuscript, we have changed the wording to highlight that body silhouettes were self-reported and based on the fathers' own perception of their body size at different time points. That being said, to distinguish between normal weight and overweight fathers, we applied the same cutoffs (overweight: body silhouette 5 or greater in men) that a previous validation study defined as being optimal for identifying overweight people (BMI, 25-30 kg/m²) (Dratva J. et al. Validation of self-reported figural drawing scales against anthropometric measurements in adults. *Public Health Nutr.* 2016;19:1944–1951. doi: 10.1017/S136898001600015X). Moreover, the figural drawing scale tool has been validated against measured and self-reported height and weight, both with regard to current (Dratva et al, see above) and past body silhouettes in this cohort (Lonnebotn M. et al. Body silhouettes as a tool to reflect obesity in the past. *PLoS ONE.* 2018;13:e0195697. doi: 10.1371/journal.pone.0195697). We believe this justifies describing fathers with body silhouette 5 or greater as being overweight. Revised explanation and the references are given in the manuscript's methods section lines 130-140.

10. The abstract starts with a strong statement that has not sufficiently been proven yet, especially in humans. “Boys’ prepubertal overweight appears to cause asthma and low lung function in future offspring,”... Causality has not been proven in human studies so far, hence the wording “cause” needs to be replaced, using “associated with”, “related to”,...

In the revised version of the manuscript, we have changed the wording to associated/related to, as we acknowledge that not all of the epidemiological reports we have cited are designed to prove causality. That being said, in the study by Lonnebotn et al. (doi: 10.3390/nu14071506) on parental prepuberty overweight and offspring lung function, an advanced statistical approach using counterfactual mediation models were used for assessing causality (Arah OA. Commentary: Tobacco smoking and asthma: multigenerational effects, epigenetics and multilevel causal mediation analysis. *Int J Epidemiol* 2018; 47: 1117–1119. doi: 10.1093/ije/dyy193). The study results were also supported by probabilistic simulations on the impact of unmeasured confounding (Lutz SM, Thwing A, Schmiede S, et al. Examining the role of unmeasured confounding in mediation analysis with genetic and genomic applications. *BMC Bioinformatics* 2017; 18: 344. doi: 10.1186/s12859-017-1749-y). So at least for this paper, we do think it is scientifically sound to refer to the study findings – that fathers’ overweight starting before puberty appears to cause lower FEV1 and FVC in their future sons.

11. Further, the abstract misses important information, such as the specific method used to identify differentially methylated cytosine-phosphate-guanine (dmCpG) sites, the fact that “body silhouette” was self-reported or a perception (or body image)... , and the tissue type used to measure dmCpGs needs to be added as well.

We appreciate these comments; the abstract has been revised to address these points.

12. Abstract shows 2 short lists of genes, but it is not clear where these came from in the text body (“top hit” is mentioned, but it is not clear what is the meaning of “top hit”). We have added a few top genes from FBS-V8c and FBS-V30c and clarified that ‘top hit’ is based on the order of FDR p-value. Similarly, we have added a few imprinting genes.

13. Inter- and transgenerational is not the same; hence, why concluding about “multiple generations”?

We agree with the reviewer that inter- and transgenerational epigenetic inheritance is not the same and we cannot prove transgenerational inheritance from this study. In the revised manuscript the wording has been changed to ‘the next generation’.

Introduction.

14. Lines 76-77: more information about studies (ref 2-8...) is needed. Are these independent studies? Other than the current one? Performed by other researchers? Association studies?...

It seems that the rationale is based on their own data and the same data set (RHINESSA) as the current one. Hence, it looks like most of this section (rationale and hypothesis) is built on data from the same research group.

However, the literature does show several related reports!

In short, the introduction is too brief and misses “supporting” references about fatherchild inheritance through epigenetic mechanisms, from other groups. At least the following (pioneering) references are missing (Soubry et al. 2015; Potabattula et al. 2019; Donkin et al. 2016; Keyhan et al. 2021; Soubry et al. 2013; Kaati, Bygren 2002; Tomar et al, 2024; Soubry 2018)

About mechanisms and inheritance (Soubry et al. 2014; Waterland and Jirtle; Noor et al. 2019)

We thank the reviewer for drawing our attention to this. In the revised manuscript we have pointed out that studies 2, 3 and 4 in the reference list are epidemiological studies, based on data collected in two different multigenerational cohorts, the RHINESSA generation study and the Tasmanian Longitudinal Health Study (TAHS). These, and other studies we are citing, are independent studies even though some of them originate from data collected from the same international population-based studies; The European Community Respiratory Health survey (ECRHS), the Respiratory Health in Northern Europe study (RHINE), and the RHINESSA generation study, which aims to investigate the offspring of participants from RHINE and ECRHS study centres. In the revised introduction we could have added more information on these study findings, however, in order not to make the introduction too lengthy, we have rather focused on incorporating suggested references and include more information on potential epigenetic mechanisms that may underlie observations from epidemiological reports. We do think that the revised introduction is now more informative

with regard to relevant studies on preconception body composition and offspring health outcomes.

It is not clear how timing of “voice break” relates to “spermatogenesis” and DNA (re)establishment of methylation marks in sperm? Please clarify.

Evidence suggests that imprinted and paternally expressed genes are involved in initiating pubertal timing (DLK1 and MKRN3), and that the regulatory switch of puberty activating gene expression from an inhibitory to an excitatory state, which activates the hypothalamic-pituitary-gonadal axis and initiates puberty, are facilitated by epigenetic processes (Dauber et al. 2017, <https://doi.org/10.1210/jc.2016-3677>; Shalev and Meamed 2020: <https://doi.org/10.1016/j.mce.2020.111031>). Thus, we do think that the pubertal transition phase may constitute a period of increased vulnerability for lifestyle influences to drive epigenetic modifications – including obesogenic environmental factors. This has also been further elaborated in the revised discussion.

Why do the authors believe that this question is important? Is there a reason why men would remember their weight/silhouette better at the time of voice break?

Our interest in investigating fathers’ perceptions of their body silhouette around the time of voice break is, to a large extent, based on our hypothesis that the period around the pubertal transition phase may be a critical exposure sensitive time period for environmental factors to trigger epigenetic processes in the germ cells, as also highlighted in the previous response. For that reason, we regard it important to investigate whether offspring of fathers who were overweight / having a body silhouette of 5 or higher at voice break had different methylation patterns compared to offspring of fathers who were normal weight, equivalent to having body silhouette figures spanning from 1-4 at voice break. As noted above in response to point 9, the validity of recall of body silhouette has been previously demonstrated. We believe this question is of public health importance as it suggests that interventions to improve health of the populations should equally focus in adolescent boys as future fathers as much as adolescent girls as future mothers.

Method and study (population)

15. Why beta-values and not M-values? Beta-values do not have a normal distribution, and they are heterogeneous, hence do not meet the assumptions of a linear regression. Please see comment 4 in response to reviewer #2

16. The manuscript misses the exploration of specific mechanisms of epigenetic inheritance (such as through imprinting). The following should be verified through a focussed additional test: association between FBS and father's DNA methylation at imprinted genes (using data bases such as the Geneimprint data base or the human Imprintome. See: We thank the reviewer for suggesting this further analysis. In the revised manuscript we have used the list of imprinted genes from <https://geneimprint.com/site/genes-byspecies.Homo+sapiens> to assess whether imprinted genes are over-represented in the list of genes that the 2005 CpGs associated with paternal body silhouette map to. This shows that 47 dmCpGs mapped to 24 genes known to be imprinted genes. Furthermore, we have also assessed for enrichment of metastable epialleles, loci whose epigenetic modifications are established during early embryonic development (Silver MJ, Saffari A, Kessler NJ, et al. Environmentally sensitive hotspots in the methylome of the early human embryo. *Elife*. 2022;11: e72031.). Our analyses found 37 dmCpGs overlap with known metastable epialleles (see supplementary Data 11). We have also included this in the results section and discussed more fully known associated traits from the EWAS atlas to improve interpretation for the reader.

17. Study population and cofactors: Age range is large (7 -51 years) for a sample size of about 300 offspring. Age is a major confounder. Did the authors include findings from other studies to correct for age?

We have used age as covariate and compared age overlap with age related dmCpGs from EWAS atlas. There are N=188 dmCpGs known to be associated with age.

18. Same issue about smoking status of offspring?

As sensitivity analysis, we have conducted EWAS where smoking status of offspring was included as a covariate (see supplementary data 1B). We showed that 19 of our dmCpG sites remain significant. Moreover, in the look-up EWAS atlas, out of 2005 dmCpGs, there is only a small overlap with mother's smoking (55) and current smoking by offspring (118), see supplementary data 9.

19. Another important potential confounder that needs to be verified is study centre. Study centre was included as covariate. As sensitivity analysis, we have run EWAS on the Bergen centre alone as the largest centre, and the results are broadly similar. See response to reviewer 2, comment 4.

20. Line 191-192, etc: How did the authors account for the fact that the study has very imbalanced classes (307 vs 32)? Only 10% of fathers had overweight. Hence, this study is based on only 32 subjects that have been “exposure”; taking into account the large age range and heterogeneity of several other (unmeasured) factors, is a major weakness of this study. Age range is large (7-51y) and data come from a very heterogenous population, from Europe till Australia.

We acknowledge as a potential weakness the limited number of exposed individuals, however, it is also a strength that our study is based on a population based generally healthy cohort, which means the results can provide knowledge about a general population rather than only a selected group of exposed and non-exposed persons. We further acknowledge, that it is important in this type of study design to include study centre and age as covariates like we have done; further, the study participants are of white European ethnic background. Even when accounting properly for this heterogeneity, we still observed significant associations. See comment #7

21. Six different study centers were included, how was the distribution of overweight, age, ...by centre? It needs an additional demographics table. Thank you, please see demographics table 1B.

22. Other weaknesses are the use of “silhouettes” using only 2 categories (in the major analysis?). FBS was divided in 2 categories, and from 1 till 7. But no further explanation is given (how were these defined? Calculated?). If published earlier, it could be included in the general figure /overview or as suppl. information.

In the revised manuscript we have tried to make it more clear how we classified fathers as being normal weight and overweight based on previously identified body silhouette cutoffs, see lines 133-135. We have included references showing that the figural drawing scale tool has been validated against measured height and weight, both regarding current and past body

silhouettes, and also that the cut offs we have applied are the same as those defined in the validation study as being optimal for identifying overweight people (BMI 25-30 kg/m²). For a more elaborate response, please see our response to comment 9.

23. It is a self-reported “estimate of BMI”. The authors refer to validation tests, but these should be added (or repeated) in suppl. data for clarity. References from other studies (performed others) are needed showing that this method is valuable. There are certainly papers on this, here is one example on reliability of a BMI-based Silhouette Matching Test (Body-image perceptions: reliability of a BMI-based Silhouette Matching Test Peterson M, Ellenberg D, Crossan S. 2023).

The suggested paper has been included in the revised version. Further, please see our response to comment 9 and comment 22. References to the validation reports from Dratva et al. and Lonnebotn. et al have been included in the manuscript, and these have several more references to the use of body silhouettes. Finally, the figure of the figural drawing scale of the 1-9 body silhouettes is provided in Supplementary figure 1.

24. Also asthma was self-reported, based on one single question: “Have you ever had asthma diagnosed by a doctor?”. It was not verified in a medical record afterwards? Self-reported physician diagnosed asthma is documented to be a very specific, but not so sensitive, measure of clinical verified asthma (Torén et al. Chest 1993; 104: 600-608; Pekkanen J, Pearce N. Eur Respir J 1999; 14: 951-957; Torén K et al. Respir Med 2002;96:635). In Scandinavian populations, self-reports have been demonstrated to have 92.4% specificity to health care records (Jensen HAR, et al. Agreement between self-reported diseases from health surveys and national health registry data: a Danish nationwide study. J Epidemiol Community Health. 2023;77(2):116-122. PMID: 36446554). Although the use of self-reported asthma is commonly used to determine prevalence of asthma in epidemiological studies, we are aware that this can introduce misclassification, which we also have addressed in previous papers of the RHINE study population (Torén, Svanes et al. on behalf of the RHINE study group. Eur Respir J 2004; 24: 942-946). Thus, the use of self-reported asthma in epidemiological studies rather tend to lead to an under-detection of subjects with asthma. In our study, some asthmatics may be misclassified into the non-asthmatic group and our results might be stronger if we had a perfect classification of asthma, however, it is likely that

our use of self-reported asthma based on the question “have you ever had asthma diagnosed by a doctor” to a large extent captures subjects with an actual asthma diagnose.

25. It would be informative to add a table on the frequencies of “asthma”, “lung function”, and frequencies of BMI (or obesity status) in offspring, by centre, by gender, ... Thank you, please see table 1.

26. Was there a significant association between socio-economic status of the grandparents and the father’s body measurements? It could be added in results or supplementary material how father’s measurements differ between socio-economic status of grandparents.

Please see supplementary data 1B.

27. In general, the number of covariates is limited. It is possible that the results are based on unmeasured factors.

Please see QC figure in comment #7. Further, regarding the limited number of covariates included in the analyses, please also see responses to comments 5 and 20. In general, we have included the variables that we consider fulfil the criteria for confounding in our analyses, but we cannot exclude the possibility for rest-confounding. This is now discussed in the revised manuscript (lines 675-83).

Results/methods

28. The presentation by each “exposure to silhouette of the father by his age” separately, is not sufficient to draw clear conclusions. Because the outcomes of DNA methylation were measured once in offspring (and hence, these methylation outcomes are the same for all exposures through the father). It is warranted to perform a comparative data analysis using all paternal information in the same model. E.g., using matrices, or using a categorization of windows of exposure, in one mathematical model (e.g., a positive change in silhouette between age 8 and voice break, versus, a positive change after voice break, one single exposure of “overweight”, being twice “overweight” in 2 windows, etc...)? If this could be done/shown, only then this sentence makes sense: Line 888: “Importantly, the period around voice break stands out as the most critical age window regarding epigenetic effects...”

As father body silhouette varies with his age, in our investigation we were interested in identifying associated epigenetic marks. In the EWASs of three time points and change of FBS during adolescence, we observed a greater signal (dmCpGs sites) around voice break as evidence for our conclusion (Supplementary data 1). For clarity for the reader we have restated the conclusions as “In particular, changes in father’s body silhouette status across voice break, from a normal weight to an overweight FBS or vice versa, were associated with altered DNA methylation patterns in the offspring.” (see line 482-484).

29. The direction and importance of changes at CpGs have not been elaborated or discussed.

Such as increased versus decreased, imprinted (hyper/hypomethylation),... For hypermethylation, see figure 2 and for imprinting, supplementary data 11.

30. Only 2 imprinted genes were identified.

Why is this not mentioned in the abstract? As it is important to explain inheritance.

B4GALNT4 in males: In table 3, explain why this CpG has 8 % methylation? (is this imprinted at the paternal allele?)

An interpretation is missing. For instance, this means that if the father had overweight at age of voice break, his son is at risk to have +2% at this site... Similar comment about FERMT1: 84% methylated in blood.

Note, “AveExpr” is confusing (see Table3), although it may be the output of the program, I assume it is “average methylation” (0 – 1; where 1 is 100%)?

Overall I miss interpretation of the data and explanation of their findings. We have updated the results including a comparison of genes mapped to dmCpGs that overlap the gene list from imprintome. See comment #16 and supplementary data 11.

Just to give some more examples:

If imprinted, how can methylation marks already established before the age of 11, become aberrantly differentially methylated after prepuberty (at end of puberty/“voice break”)? The correlation of imprinted dmCpGs with age was weak as shown in supplementary data 11, in age tab.

31. How can a small effect in sperm (small % of sperm cells) of 32 fathers result in a measurable change in DNA methylation of blood cells in a small number of children? (32 from overweight fathers, e.g., 16 were boys and 16 were girls)?

While the precise mechanisms by which paternal environmental exposures effect the phenotype change in the offspring are not fully established. It is clear that, at least for some exposures, this involve alterations in small non-coding RNA (sncRNA) content of spermatozoa (e.g. see Yin X, et al. Paternal environmental exposure-induced spermatozoal small noncoding RNA alteration meditates the intergenerational epigenetic inheritance of multiple diseases. *Front Med.* 2022;16(2):176-184. PMID: 34515940). However, to date, to our knowledge, no single-cell analysis of spermatocytes has been undertaken in these models, thus it is not possible to state if changes in sncRNA content is consistent across all spermatocytes or only evident in a % of spermatocytes.

32. Lines 464: sensitivity test results need to be added in suppl. tables.

See supplementary table 13.

Discussion:

33. This is overstated and not correct:

(line 358):“In this first study of DNA methylation in humans as related to father’s overweight/obesity during stages of sexual maturation,...” and line 456: “to our knowledge, this is the first EWAS to use father body silhouettes in humans to identify obesity-related markers”... line 460 “that there are no other cohorts with detailed data on overweight during fathers’ preconception lifespan, that allow replication of our findings”...

See other literature where obesity (etc.) in puberty and other periods of timing during spermatogenesis and development of sperm have been studied. Calling “voice break” (instead of “puberty”), or “silhouette” (instead of “overweight”, in other studies), does not make the current study more unique than earlier studies. Instead, the current study includes self-reported data... while earlier studies included measured data (BMI ...). Although this study is not unique, consistency in the results is important to show causality. Hence, as stated in their first sentence (“cause of”), there is a need to do comparative analyses with earlier similar publications.

Please see supplementary data 14. Further, in the revised version of the manuscript, we have changed the wording of these sentences. We have also compared our results with other

studies on paternal preconception obesity and DNA methylation changes in sperm samples and offspring cord blood.

34. Line 457: “about 70% of the genes identified by the body silhouette related dmCpGs signify novel associations with obesity, not previously captured using BMI measurements.” Please clearly refer to the table where this (70%) is presented.

The statement has been updated (line 472-76) as follows:

We conducted overlap look-up for our 1261 genes (mapped from our 1962 dmCpGs) with obesity-related known genes in open targets (EFO_001073) (n=5884) and DisGeNET (N=2820) (accessed on 16/07/2022). We identified overlap with 450 genes in either of the two databases while 171 genes were reported by both databases. Our 811 genes (64.3% of 1261) are new reports (see Supplementary Figure 3 and Supplementary data 14).

35. Line 893: “smoking before age 15 years”, reference 1 was not the first showing this association. add references:

Prepubertal start of father’s smoking and increased body fat in his sons: further characterisation of paternal transgenerational responses (Northstone et al. 2014).

The reference has been included as suggested.

36. A general remark: just as in the introduction section, the discussion misses a thorough comparison of these findings with findings published by others on similar exposures and outcomes. As said, published data bases could be used to verify potential overlap or similarity in the results.

All recommended literature has been included in the introduction and referenced in the discussion. Additional analysis on reproducibility was conducted, including Bergen centre only, beta-value vs M-value EWAS at voice break for sensitivity, enrichment for imprinting genes, DMR regional pathway enrichment, overlap of known obesity genes (see earlier responses to comments).

Minor comments

37. Line 76-77 or in line 92: add average known/expected age (or range) of “voice break”. In a Scandinavian population, voice break has been demonstrated to occur at mean age 13.6 (95% CI: 13.5–13.8) years (Busch AS, et al. Voice break in boys-temporal relations with other pubertal milestones and likely causal effects of BMI. Hum Reprod. 2019;34(8):15141522. PMID: 31348498). In this study, voice break correlated moderately strongly with timing of male pubertal milestones, including testicular enlargement, gonadarche, pubarche, sweat odor, axillary hair growth and testosterone above limit of detection (r^2 range: 0.43–

0.61).

38. Is the format of the reference list correctly implemented? Some papers are indicated as 1 author et al., others are fully mentioned (including all coauthors).

The referencing has been re-formatted following Nature style.

39. Check and review figure 1 and Supplementary Figure 1 for their scale, clarity, references, scale measures of gradation in body size changes for defining changes in FBS.

Figure numbering has been updated.

40. Line 157-158: does the FDR are multiple testing adjustment refers to the covariates? It is not clear.

All FDR results are based on covariate adjusted model with $FDR < 0.05$ cut off. This is clarified in the methods (see lines 166-69).

41. Line 161 etc: Not clear, it seems like offspring was measured at multiple time points...?

Father’s body silhouette was measured at multiple time points, which is now more clearly described in lines 129-40 and 163-66.

42. Results. Father’s FBS at voice break. Line 213 (same line 225 and throughout the text:

“top-hit”): “top-10 dmCpGs” please clarify (based on p-value, magnitude, both, ...? A statement has been added to clarify that this is based on order of FDR.

43. Results. Change in FBS between 8 and voice break. Line 231: explain $\lambda=1.05$? (is not mentioned in methods section).

Thank you, now explained in line 188.

44. Results. DMRs analysis. Why are the number of differentially methylated regions much lower than CpGs; explain in the discussion.

DMR region represented by 2 or more CpG sites. We have also used DMRCate and reproduced many genomic loci as DMRs (see supplementary Data 2 and see line 308- 314).

45. Results. Offspring's current BMI and current body silhouette. Line 345: "We found 182 dmCpGs ... and 1 dmCpG associated with offspring body silhouette", in suppl. Figure 1 shows they are around 100 with 33 overlapping the one of offspring BMI. In general, numbers do not correspond between the figure and text. Where is supplementary figure 2?

Line 346: "the top hit for both..." Both which?

From our total FBS related 2005 dmCpGs, 1962 were unique. We have updated the list, and the overlap is also provided in supplementary Data 13 and supplementary Figure 2.

46. Discussion. Line 388-389: "Many observational studies obesity traits and BMI", reference? Line 410: "shorter telomere length is known to be associated with childhood obesity and BMI.", ref? Line 419-420: "Six dmCpGs were located to NUP210L ... ", ref? In general, when it is reported "it has been shown, it is known, .." a reference is needed.

In the revised version of the discussion, appropriate references have been included.

47. Results. Line 198: "strongest effect"? Larger? Reference beta difference.

To clarify, the term 'effect size' has been used.

48. Results. Line 248: misleading title, it seems DMR are found in the father. Maybe specify "offspring DMR".

Thank you, throughout the revised manuscript we are now referring to offspring DMR.

49. Results. Enrichment for gene ontology terms. Line 294: "Largest GO terms" or the one that appears the most?

This means number of GO terms.

50. Figures: Figure 2: Clarify why sometimes CpG sites names are added and some other gene names are added. Also, A and B labels are missing in the graph, so either add it or change in the legend to the right and left graphs.

CpG name used if dmCpGs are from an intergenic region. For clarity for the reader, this included in the figure 3 caption.

51. Figure 3: Clarify why sometimes CpG site names are added, and some other gene names are added.

CpG names are used for intergenic dmCpGs

52. Figure 4: Add a definition of the size of the dots to be more precise, either as a legend in the graph or in the legend of the figure as text.

We have added text legend to Figure 4 'dot size corresponds to level of correlation'.

53. Figure 6: In the A graph no genes names appear in the graph, whereas they do in the B graph. Gene names have been added to graph A

54. Supplementary discussion. Line 982-986: change font.

The font has been changed.

55. Suppl. Figure: not clear. Offspring body silhouette and offspring BMI are outcomes, while EWAS 2005 is? This is not explained in the legend.

Thank you, this has been revised, see comment #45